METHODS AND RESOURCES

# A cost-effective and scalable barcoded library construction method for deep mutational scanning studies

Jessica Jann[1,2,3,4,5,6], Isabelle Gagnon-Arsenault[1,2,3,4,5,6], Alicia Pageau[1,2,3,4,5,6], Alexandre K. Dubé[1,2,3,4,5,6], Anna Fijarczyk[1,2,3,4,5,6], Romain Durand[1,2,3,4,5,6], Christian R. Landry [1,2,3,4,5,6]*

1 Département de biochimie, de microbiologie et de bio-informatique, Faculté des sciences et de génie, Université Laval, Québec, Canada, 2 Institut de biologie intégrative et des systèmes, Université Laval, Québec, Canada, 3 PROTEO, Le regroupement québécois de recherche sur la fonction, l'ingénierie et les applications des protéines, Université Laval, Québec, Canada, 4 Centre de recherche sur les données massives, Université Laval, Québec, Canada, 5 CRI, Centre de recherche en infectiologie, Université Laval, Québec, Canada, 6 IID, Institut intelligence et données, Université Laval, Québec, Canada

* Christian.landry@bio.ulaval.ca

## Abstract

Recent developments in DNA synthesis and sequencing allow the construction of comprehensive gene variant libraries and their functional analysis. Achieving high-replication and thorough mutation characterization remains technically and financially challenging for long genes. Here, we developed an efficient, affordable, and scalable library construction approach that relies on low-cost DNA synthesis and standard cloning technologies, which will increase accessibility to mutational studies and help advance the field of protein science. Each degenerate codon variant is physically associated with multiple DNA barcodes during synthesis, which overcomes the need for long-read sequencing for linking variants to barcodes. We demonstrate the scalability of our approach by constructing a complete library for the multidrug resistance gene *PDR1*, a 3.2 kb gene encoding a pleiotropic transcription factor in the yeast *Saccharomyces cerevisiae*. We demonstrate a near-perfect correspondence in the measurement of amino acid variants impact when assessed by barcode sequencing and direct sequencing of the mutated coding sequence.

## Introduction

Understanding how genetic variation alters protein function is a central goal in genetics, molecular biology, and evolutionary and biomedical research. Over the past decade, deep mutational scanning (DMS) has emerged as a powerful technology to systematically evaluate the functional consequences of all possible single amino acid substitutions in a protein within a single experiment [1,2]. By coupling a comprehensive variant library, selection assays, and high-throughput sequencing, DMS allows

**Data availability statement:** All data are available in the main text or in the Supporting information files. The numerical data underlying all figures are provided in the files S1 Data (corresponding to Fig 1B), S2 Data (corresponding to Figs 1D, S2, S4, S6, S8, and S9), S3 Data (corresponding to Figs 1C, S1, S3, S5, S7, and S9), S4 Data (corresponding to S10 and S11 Figs), S5 Data (corresponding to Figs 2B and S13), S6 Data (corresponding to Figs 2B and S13), S7 Data (corresponding to Fig 2C), S8 Data (corresponding to S12B Fig) and S9 Data (corresponding to S14 Fig). Sequencing data are available in the NCBI Sequence Read Archive (SRA) under BioProject PRJNA1269095. All code and processed data to reproduce the analyses, results, and figures are available at https://github.com/Landrylab/Jann_et_al_2025 and archived on Zenodo (DOI: https://doi.org/10.5281/zenodo.18342596).

**Funding:** This work was supported by a Genome Québec and Genome Canada grant (6569), a Canadian Institutes of Health Research (CIHR) Foundation grant (387697), a CIHR project grant (202409PJT), and a Natural Sciences and Engineering Research Council of Canada (NSERC) CREATE grant (EvoFunPath) to CRL. JJ and RD are supported by postdoctoral fellowships from NSERC. JJ benefits also from FRQS postdoctoral fellowship (343898). CRL holds the Canada Research Chair in Cellular Systems and Synthetic Biology. The funders had no role in study design, data collection and analysis, decision to publish, or preparation of the manuscript.

**Competing interests:** The authors have declared that no competing interests exist.

**Abbreviations:** AMP, ampicillin; DMS, deep mutational scanning; DMSO, dimethyl sulfoxide; DOX, doxycycline; G418, geneticin; ITR, itraconazole; mEGFP, monomeric enhanced Green Fluorescent Protein; NAT, nourseothricin; POSA, posaconazole; SC, synthetic complete; WT, wild-type.bp, base pair; PDR1, Pleiotropic Drug Resistance 1; YFG, Your Favorite Gene.

the systematic quantification and mapping of genotype-phenotype relationships at high resolution and coverage [1,2].

DMS involves four major steps: (i) the construction of a comprehensive variant library, typically introducing all possible single amino acid substitutions into a target gene (or specific locus); (ii) the delivery of this library into a biological system via vectors (such as plasmids) or genomic integration; (iii) the application of a selective pressure (e.g., drug exposure, receptor activation, sorting-based protein stability); and (iv) the quantification of variant frequencies before and after selection using deep sequencing, allowing the estimation of functional effects for each substitution. Connecting mutations to phenotypes leads to the construction of mutation-phenotype landscapes that provide insights into diverse biological processes, such as structure-function relationships [3], disease-associated mutations [4], and mechanisms of adaptation [5].

The robustness and completeness of a DMS experiment rely largely on the quality and design of the variant library. Constructing a systematic variant library offers several advantages over random or non-systematic approaches (e.g., error-prone PCR) [6,7]. It ensures exhaustive mutational coverage and removes biases introduced by targeted or region-focused mutagenesis. Furthermore, vector-based variant libraries offer flexibility in downstream applications: they can be easily stored and reused in different genetic backgrounds or biological systems [8], coupled with fluorescent tags or reporters [9], and they allow flexible expression under constitutive or inducible promoters [10]. Vector-based libraries can be used not only for transformation but also as templates for transferring mutations into genomic loci using CRISPR-Cas9 editing or homologous recombination [11,12]. Examples of vector-based DMS applications include signaling-specific variants of MC4R in obesity [13], immune escape mutations in the SARS-CoV-2 Spike protein [14], and pathogenicity classifications for CRX and PAX6 in inherited eye disorders [15]. In microbes, recent efforts have used DMS to map resistance-conferring mutations [5,16,17].

While DMS is a powerful tool for high-throughput functional analysis, generating reproducible and high-coverage data remains challenging. Incorporating internal replicates, such as multiple unique codons or DNA barcodes per amino acid variant, reduces technical noise, improves the precision of effect estimates, and enhances sensitivity and reproducibility [18–20]. The use of plasmid-based libraries further facilitates linking each variant to unique short DNA barcodes, enabling cost-effective tracking of mutation frequencies across diverse conditions using short-read sequencing. Similar barcoding strategies have previously been used in CRISPR-based pooled screens to enable high-throughput variant tracking [21].

Although the cost of high-throughput sequencing has steadily declined over the past 20 years, the large size of many target genes makes the cost of such experiments prohibitive. Gene size limits DMS scalability due to the complexity of the variant library and the associated synthesis and sequencing costs, including for linking variants to DNA barcodes. To overcome these challenges, we developed an efficient and cost-effective two-step cloning strategy for plasmid-based DMS libraries that facilitates the study of large genes using DNA-barcoded synthetic oligonucleotide variants.

Our approach involves fragmenting a gene of interest (*Your Favorite Gene: YFG*) into subregions of up to 150 base pairs (bp), in accordance with the synthesis length limits of single-stranded oligonucleotide pools (oPools). The oPools are composed of five parts (Fig 1A): (i) ~40 bp of homology upstream of the *YFG* fragment of interest, (ii) a *YFG* fragment sequence containing a single NNK codon substitution, (iii) BsaI restriction sites, (iv) a 30 bp DNA barcode alternating random dinucleotides (NN) with codon position-specific sequences, and (v) a conserved i7 primer binding site (PBS_i7).

The integration of a barcode facilitates variant identification while allowing each codon, and thus each amino acid substitution, to be represented by multiple distinct barcodes, thereby providing internal replicates for downstream analysis. The barcodes and their corresponding variants remain physically linked on the same DNA molecule (plasmid), ensuring accurate tracking throughout the experiment.

This strategy utilizes the economical oPools, which are ~100-fold cheaper per base than double-stranded DNA synthesis (<$0.001 versus ~$0.05 per base) [22,23]. In addition, low-cost short-read sequencing is used (i) to associate random barcodes to their corresponding coding sequence variants and (ii) to track mutations in downstream experiments via the barcodes. This approach offers a cost-effective alternative to methods that require long-read sequencing for variant identification and tracking [24,25].

The cloning strategy is divided into two steps: a Gibson assembly, followed by a Golden Gate assembly (Fig 1A). In the first step, the plasmid backbone is generated via PCR from a yeast expression vector containing the wild-type (WT) *YFG*. Gibson assembly allows inserting the oPools (variant fragment + barcode + PBS_i7) into the backbone. These constructs then serve as backbone plasmids for the second cloning step.

In the second step, the missing *YFG* 3′ end sequence is amplified, including the C-terminal coding region, terminator, and PBS_i5. Golden Gate assembly enables the one-pot assembly of this missing *YFG* 3′ end sequence into the Gibson-assembled plasmids, using BsaI digestion and vector-insert ligation cycles (see Methods section). This strategy results in one distinct plasmid library per *YFG* fragment, each containing a full-length *YFG* sequence carrying a single mutation (in the targeted fragment) specifically linked to a DNA barcode flanked by the PBS_i5 and PBS_i7.

Each library undergoes quality control by high-throughput sequencing at both cloning steps (Fig 1):

- After Gibson assembly, a necessary short-read sequencing (mutated fragment + barcode) is used to (i) establish barcode-mutation associations and (ii) assess both mutation coverage and barcode diversity per mutation.

- After Golden Gate assembly, only the DNA barcodes are sequenced (facultative) to confirm that coverage and diversity have been maintained.

Our method is unique but shares some key elements with other methods previously developed, while being significantly more affordable. For instance, Jones and colleagues [26] (and slightly modified by Howard and colleagues [27]) also employed a two-step approach to link barcodes to variants, thereby overcoming the need for long-read sequencing. Their approach relies on pools of individual oligonucleotides with the specific sequences to be screened, i.e., without degenerate positions. The variants are linked to a random DNA barcode (15N) following their synthesis using a degenerate oligonucleotide, and this occurs before the missing gene fragment is inserted. There are major differences with our approach. The gene variant synthesis relies on more expensive gene synthesis technology. Second, the approach utilizes a random barcode, which increases the complexity of the library and potentially increases the fraction of low-complexity barcodes that can cause issues during downstream sequencing. Our design enables a library with a codon-specific barcode, which helps reduce sequence complexity and allows tracking positions and fragments easily for quality control. This is made feasible because the barcodes and mutated gene fragments are initially located on the same DNA molecules. An approach that allows for the use of affordable oPools like ours is SUNi mutagenesis [28]. This protocol employs a completely different method for mutagenesis, using degenerate pools of oligonucleotides that anneal to single-stranded plasmids to introduce variants. One noticeable difference between this approach and others is the proportion of mutants successfully recovered. While our method achieves near-complete mutation coverage (99.8%), the authors report substantially lower

 

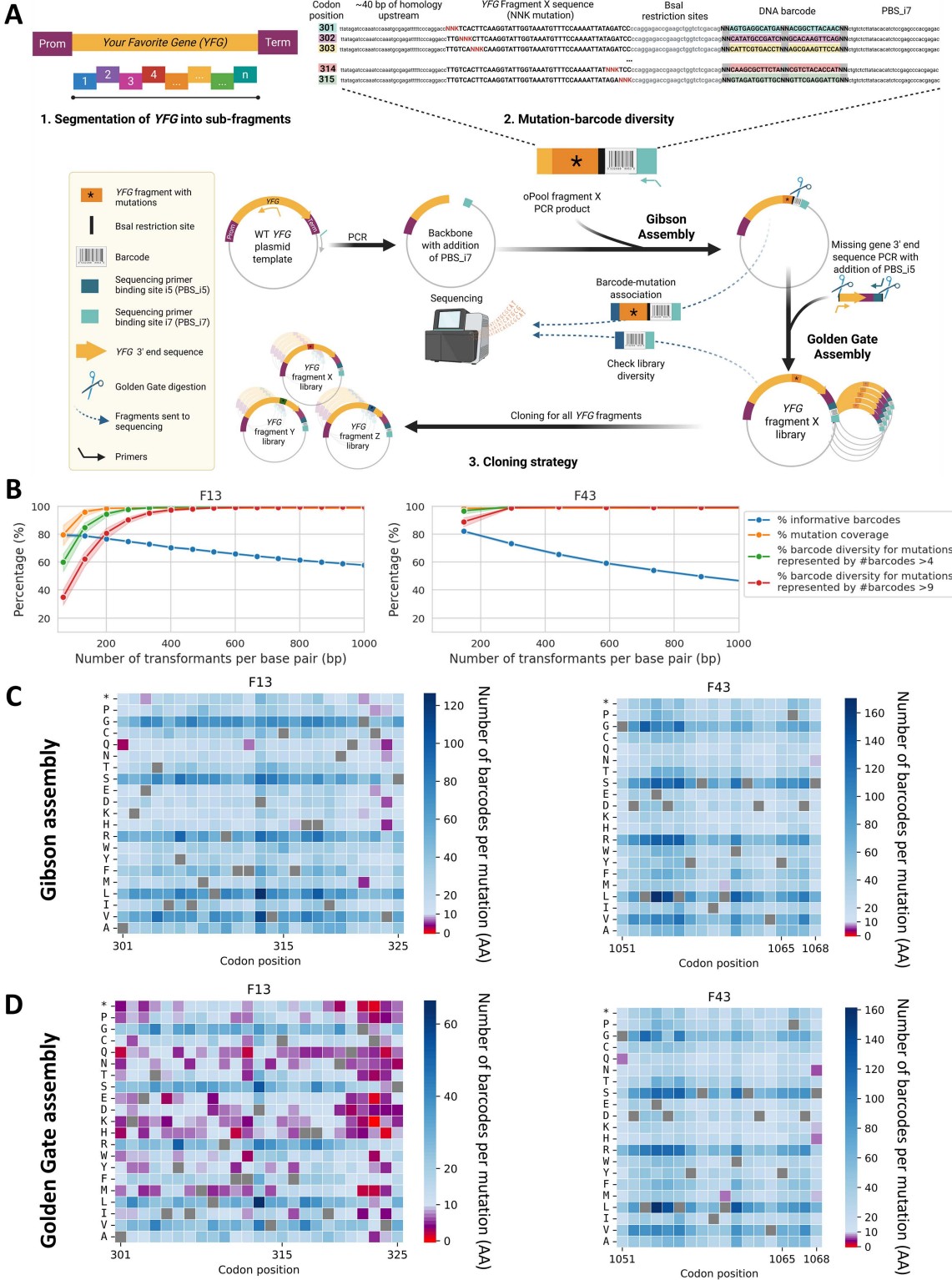

**Fig 1. A) Strategy to generate a DMS plasmid library for *Your Favorite Gene* (*YFG*) using short, degenerate libraries. 1.** Segmentation of *YFG* into sub-fragments, each fragment corresponding to a DNA region to be synthesized. The same approach can be applied to promoter and terminator regions, if desired. **2.** Example of a pool of degenerate oligonucleotides (oPool) derived from one *YFG* fragment associated with DNA barcodes. Each

oPool contains: (i) ~40 bp of homology upstream of the *YFG* fragment of interest, (ii) the *YFG* fragment sequence with a single NNK codon, (iii) BsaI cloning sites, (iv) a DNA barcode composed of codon-position specific regions and six degenerate nucleotides (N), and (v) a conserved i7 primer binding site (PBS_i7) present in all oPools and used for rapid and efficient sequencing library preparation. Current oligonucleotide synthesis technologies allow for a total of nine degenerate positions per fragment: three are used for the degenerate codon (NNK), and six for the barcode. A complete list of all oPool sequences and their detailed composition is provided in S1 Table. **3.** Protocol for constructing *YFG* DMS plasmid library from oPools using two cloning steps that maintain the physical barcode-mutation association. The libraries of oPools are cloned into the plasmid template by Gibson cloning. Following this step, for each fragment, a necessary short-read sequencing using PBS_i5 (included in the 5′ sequencing primer) and PBS_i7 is performed to associate each barcode with its corresponding mutation and to assess both barcode diversity per mutation and mutation coverage for the whole fragment. The ultimate step consists in Golden Gate cloning of the missing 3′ gene fragment between the degenerate fragment and the barcode. An additional short-read sequencing step of the barcodes can be performed to make sure that coverage and diversity have been maintained. Figure created in BioRender. Barff, T. (2025) https://BioRender.com/1vfl2on. **B) Optimization of cloning steps.** Cumulative percentage of informative barcodes (those associated with a single coding mutation and not with the wild-type (WT)), mutation coverage (percentage of mutations represented by at least one informative barcode) and barcode diversity (percentage of mutations represented by informative #barcodes >4 or >9) as a function of the number of transformants recovered after Gibson cloning for two fragments (F13 and F43) of a long gene coding for a transcription factor. Data shown here are derived from the combined results of multiple independent small-scale transformations, and results are normalized to the number of transformants per base pair (bp) to facilitate comparison across gene fragments of different lengths. Means and confidence intervals were obtained from 100 random subsamplings of independent transformation experiments, each consisting of 5,000 (F13) or 7,500 (F43) transformants, cumulatively combined. The numerical data underlying these graphs is provided in the file S1 Data. **Library quality control after (C) Gibson and (D) Golden Gate assemblies for PDR1 F13 and F43.** Heatmaps show barcode diversity for each possible amino acid substitution at each codon position. For each fragment, a total of 25,000 and 100,000 transformants were recovered and analyzed after Gibson and Golden Gate assemblies, respectively. Barcode diversity is shown using an unclipped color scale, allowing visualization of the full range of barcode counts. Mutations covered by high barcode diversity (#barcodes ≥10 and ≥4) are represented by blue and purple scales, respectively, while lower barcode diversity (#barcodes <4) is represented by a red scale. Gray squares represent WT amino acids. The numerical data underlying these graphs is provided in the files S2 and S3 Data. Clipped versions of these graphs (maximum of 10 barcodes per mutation), optimized to highlight lower barcode diversity, are provided in S1 and S2 Figs.

coverage, with 77% and 69% of mutation coverage for two independent libraries. In addition, this protocol does not use DNA barcodes that bypass the need for tile-sequencing, particularly for long genes, although it could arguably be adapted to do so. Finally, DIMPLE [29] relies on oligonucleotides generated using DNA microarrays, which likely improves coverage compared to SUNi, but at a significantly higher synthesis cost. One similarity between the DIMPLE approach and ours is the use of Golden Gate cloning, specifically for assembling the synthesized mutated gene fragments with full-length gene backbones. The advantage of DIMPLE is the introduction of other types of mutations, such as insertions and deletions (indels). We did not consider them in our current library, but we have previously successfully used oPools for making indel libraries [30], so our protocol could also include such mutations. In addition to synthesis cost, one significant difference between our protocol and DIMPLE is that it does not allow for rapid and affordable library barcoding during gene fragment synthesis.

To validate the feasibility and the strength of our approach, DMS libraries for all 43 fragments of a long gene coding for a transcription factor were constructed and characterized (Figs 1B–1D and S1–S8). Even if the information on all fragments is provided, we focus here on fragments F13 and F43 because they cover different distances from the DNA barcodes and because F13 contains mutations of interest. The performance of a DMS plasmid library depends on four key parameters: Informative barcodes (i.e., barcodes associated with only one coding mutation and not with the WT), which ensure data usability and limit sequencing costs of non informative barcodes; Mutation coverage (i.e., percentage of mutations represented by at least one informative barcode); Barcode diversity (i.e., percentage of mutations represented by informative #barcodes >4 or >9), which provides internal replicates; Variant uniformity (i.e., how evenly individual variants are represented within the library), ensuring balanced abundance and accurate quantitative comparisons.

These parameters are largely dependent on the number of recovered transformants after each cloning step. To better estimate this relationship, we combined sequencing data obtained from multiple independent small-scale transformations and normalized the results to the number of transformants per bp, allowing direct comparison across fragments of different lengths. A random sampling of resulting distributions indicates that after Gibson assembly, ~350 transformants per

bp are needed to optimize sufficient informative barcode, mutation coverage, and barcode diversity(Fig 1B). This level of recovery yields 300–350 informative barcodes covering all possible mutations, with over 99.8% of mutations represented by #barcodes >4 and over 97.0% by #barcodes >9 (Figs 1B, 1C, and S1). Altogether, the findings demonstrate excellent mutation coverage and barcode diversity, after the first cloning step, two essential features for the robustness and reproducibility of DMS data.

As each cloning step inevitably results in some diversity loss, the number of transformants recovered after Golden Gate assembly was increased 4-fold (~1,400 transformants per bp). Although a small subset of mutations showed lower representation, our cloning strategy ensured over 99.8% of mutation coverage and preserved high barcode diversity: over 93.6% of mutations were still covered by #barcodes >4, and over 68.6% by #barcodes >9 (Figs 1D and S2).

To complement the barcode diversity and coverage analyses, the full distributions of barcode counts per amino acid mutation after Gibson and Golden Gate assemblies are shown in S9 Fig, providing a direct visualization of barcode representation across variants for fragments F13 and F43. In addition, the uniformity of variant representation was assessed after both the Gibson and the Golden Gate steps using the Gini coefficient, an index of distribution bias ranging from 0 (perfect uniformity) to 1 (maximum bias) [31]. We observed that the Golden Gate assemblies showed a slight increase in the Gini coefficient compared to the Gibson assemblies (S10 and S11 Figs), reflecting a minimal bottleneck related to multi-fragment cloning. Overall, uniformity remained high across all 43 fragments (Gini coefficients consistently <0.5).

To demonstrate the feasibility and utility of our library construction strategy, we focused on antifungal resistance, an increasingly pressing issue due to the rising incidence of invasive fungal infections and the limited number of effective treatment options [32]. Drug resistance mutations significantly compromise the efficacy of antifungal therapies, underscoring the urgent need for innovative approaches to investigate and counteract resistance mechanisms [33]. Among current antifungal drugs, azoles are widely used in clinical settings, including posaconazole (POSA), a broad-spectrum triazole [34–37]. The transcription factor Pleiotropic Drug Resistance 1 (Pdr1) plays a key role in mediating multidrug resistance [38]. Even if it regulates many downstream pathways, *PDR1* itself is not essential for yeast viability. However, mutations in *PDR1* lead to overexpression of downstream targets, including ABC efflux pump genes like *CDR1*, *CDR2*, and *SNQ2*, conferring broad-spectrum drug resistance in both the model yeast *Saccharomyces cerevisiae* and the human pathogen *Nakaseomyces glabratus* (*Candida glabrata*) [39,40]. Although numerous studies have demonstrated the central role of *PDR1*, the precise identity and functional impact of resistance-conferring mutations remain largely unknown. Thus, our cloning strategy targeted *S. cerevisiae PDR1*, a large protein of 1,068 amino acids, whose comprehensive mutagenesis would have been financially and technically challenging using conventional methods. To demonstrate the accuracy of our method, we focused on fragment 13 (F13; residues 301–325) of Pdr1. This fragment combines both unknown and well-characterized regions associated with drug resistance [38], including previously identified and experimentally validated mutations [40–44].

The *PDR1* F13 variant library was used to perform high-throughput functional screening of all possible amino acid substitutions, linking genotype to antifungal resistance phenotypes (Fig 2). The DMS experiment consisted of four main steps (Fig 2A): (i) generation of the *PDR1* variant library by transforming the plasmid library into a *S. cerevisiae* strain with doxycycline-repressible genomic *PDR1* expression (*ScPDR1*-DOX), (ii) competition assay with POSA, (iii) high-throughput sequencing, and (iv) fitness analysis of each variant.

In this system, the endogenous *PDR1* locus is placed under the control of a doxycycline-repressible promoter, allowing conditional regulation of its genomic expression. Although *PDR1* is not essential for yeast viability, its activity strongly influences azole resistance phenotypes. The design allows conditional inhibition of genomic *PDR1* expression and functional complementation with the variant libraries generated in this study, as shown in S12 Fig. With this system in place, we evaluated the performance of barcode-based sequencing for accurate quantification of variant fitness.

In principle, only short-read sequencing of DNA barcodes would be required. Here, we also performed full-length sequencing of the F13 fragment to compare the accuracy of the selection coefficients estimated from the barcode and

from the mutated fragment directly, as is routinely done for small genes [45–47] or through tile-seq for longer genes [5,48]. The estimates from the barcodes or from the F13 coding region were strongly correlated ($r = 0.97$, all available barcodes, S13 Fig). This high level of concordance was also observed with a much smaller number of barcodes (four barcodes per mutation ($r = 0.91$)), confirming the robustness of our cloning and barcoding strategy (Figs 2C and S14). Only a few discrepancies between barcode- and sequence-based estimates were observed (8 out of 525 mutations), most of which involving mutations supported by few barcodes (≤5) and showing neutral-to-sensitive fitness effects. By analyzing selection coefficients across all amino acid substitutions, we identified two distinct regions within fragment F13 with different contributions to resistance (Fig 2B and 2D). Residues 301–312 showed frequent gain-of-resistance mutations and residues 313–324 appeared to play a limited role in resistance, with fewer substitutions conferring resistance. These results closely match resistance-conferring mutations reported in the literature for Pdr1 in *S. cerevisiae*, *S. paradoxus,* and the closely related species *N. glabratus* (Fig 2B; e.g., T304N, I307S, M308I, R310S, etc.), supporting the power of our cloning strategy for mapping resistance landscapes.

In summary, we developed an efficient, scalable, and cost-effective two-step cloning strategy for the construction of high-quality, barcoded DMS plasmid libraries for genes of any length. By combining affordable oPools, DNA barcodes, and optimized Gibson and Golden Gate assemblies, while using short-read sequencing to minimize financial costs, our method ensures high mutation coverage and barcode diversity that provide robust internal replicates, key parameters for accurate and reproducible measurements of mutational effects on fitness or resistance. This approach reduces both technical and financial barriers, facilitating its broader application across diverse gene sizes, biological systems, and experimental contexts. By supporting comprehensive genotype-phenotype analyses, our strategy strengthens functional genomics and mechanistic insights, and enhances the accessibility of these technologies across laboratories.

## Methods

### General information for plasmid and strain constructions and media

All cloning procedures were performed using *Escherichia coli* strain MC1061, unless otherwise stated that we used *E. coli* NEB 5-alpha Competent Cells. *Saccharomyces cerevisiae* strain R1158, described in [49], served as the background for generating the doxycycline-repressible *S. cerevisiae PDR1* strain (*ScPDR1*-DOX, details below).

All antimicrobials were prepared as 1,000× stock solutions: ampicillin (AMP), doxycycline (DOX), geneticin (G418), and nourseothricin (NAT) were dissolved in sterile water, filtered, and stored at −20 °C; posaconazole (POSA) and itraconazole (ITR) were dissolved in dimethyl sulfoxide (DMSO) and also stored at −20 °C.

For bacterial liquid cultures, *E. coli* was grown in LB medium (0.5% yeast extract, 1% tryptone, 1% NaCl) at 37 °C with shaking at 250 rpm. For solid cultures, cells were grown on 2YT agar plates (1% yeast extract, 1.6% tryptone, 0.5% NaCl, 0.2% glucose, 2% agar) and incubated at 37 °C. When required, AMP was added at a final concentration of 100 μg/mL for plasmid selection. *S. cerevisiae* was cultured in YPD medium (1% yeast extract, 2% tryptone, 2% glucose) at 30 °C with shaking at 250 rpm. For solid cultures, 2% agar was added to YPD. Yeast transformants with pRS31N plasmids were selected on YPD plates supplemented with 100 μg/mL G418 and 100 μg/mL NAT.

All PCRs were performed using KAPA High-Fidelity HotStart polymerase unless otherwise specified. Strains, reagents, plasmids, computational tools, and additional resources are listed in S2 Table. Oligos are listed in S3 Table, and PCR conditions in S4 Table.

All transformations were performed using *E. coli* NEB 5-alpha Competent Cells (High Efficiency) following the manufacturer's recommended heat-shock protocol. Briefly, 2–5 μL of assembly reaction (Gibson or Golden Gate; see below) was added to 50 μL of chemically competent cells, followed by incubation on ice for 30 min, a 30-s heat shock at 42 °C, and immediate cooling on ice for 5 min. Cells were then recovered in 450 μL of SOC medium (0.5% Yeast Extract, 2% Vegetable Peptone, 0.5% Yeast Extract, 10 mM NaCl, 2.5 mM KCl, 10 mM $MgCl_2$, 10 mM $MgSO_4$, and 20 mM glucose) and

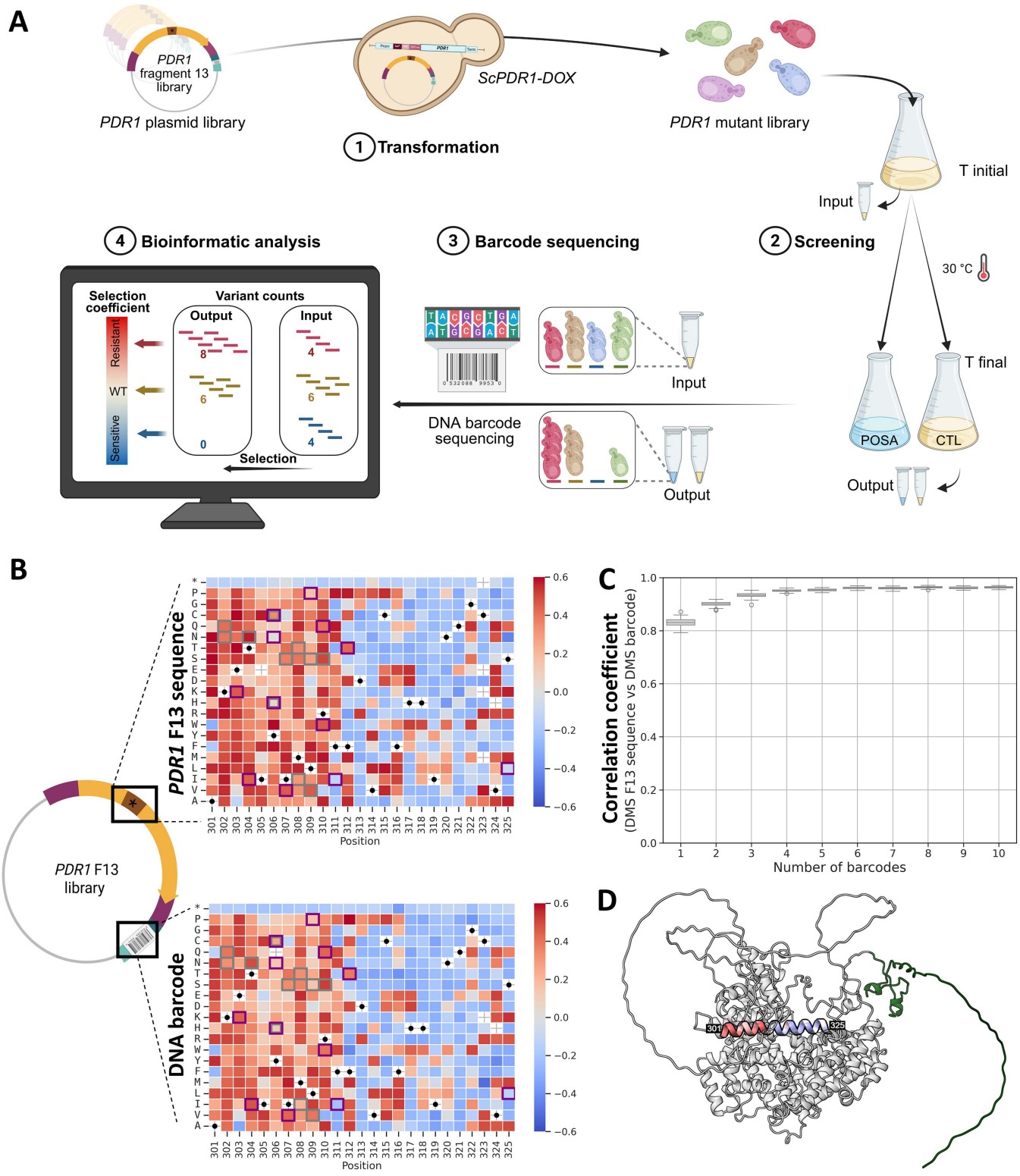

**Fig 2. Demonstration of the high-accuracy mutation profile obtained through barcode sequencing of a DMS plasmid library built from oPools. A) Pooled competition workflow using our barcoded DMS plasmid library targeting a drug resistance hotspot. 1.** The *Saccharomyces cerevisiae PDR1* variant library for F13 (residues 301-325) was transformed into a *S. cerevisiae* strain with a doxycycline-repressible endogenous *PDR1*

expression (*ScPDR1-DOX*). **2.** Transformed strain expressing the variant library was grown in the presence of posaconazole (POSA, 0.40 µg/mL) or under drug-free control (CTL) conditions. **3.** High-throughput short-read sequencing of DNA barcodes. **4.** Selection coefficient analysis to identify resistant variants. Figure created in BioRender. Barff, T. (2025) https://BioRender.com/4wpf522. **B) Azole resistance landscape of Pdr1 F13.** Heatmaps of selection coefficients for the 525 single amino acid variants screened, based on either direct sequencing of the mutated F13 region (top) or DNA-barcode inference (bottom). Positive selection coefficients (>0, red) indicate increased fitness relative to the WT, while negative values (<0, blue) indicate reduced fitness. Black dots mark WT amino acids, and asterisks represent stop codons (as internal controls). Each value is the median of synonymous codons per amino acid. White cells indicate missing data. Squares in gray (*S. cerevisiae* or *S. paradoxus*) and purple (*N. glabratus*) highlight azole resistance mutations previously reported. The numerical data underlying these graphs is provided in the files S5 and S6 Data. **C) Distribution of correlation coefficients between the selection coefficients measured with direct sequencing of the mutated F13 coding region and DNA-barcode inference**, as a function of the number of barcodes per variant considered. Each boxplot summarizes 100 random subsamplings per variant, performed using a standardized resampling procedure to normalize barcode counts across variants. The numerical data underlying this graph is provided in the file S7 Data. **D) Pdr1 tertiary structure with DMS azole resistance landscape on F13 region.** Predicted tertiary structure of *S. cerevisiae* Pdr1 (AlphaFold 3 prediction, pTM = 0.72) shown in gray with its $Zn_2Cys_6$ DNA binding domain (residues 1–78) in green. The median selection coefficients from the DMS are mapped onto F13 (residues 301–325).

incubated at 37 °C for 60 min with shaking (250 rpm) before plating on selective agar plates. The complete transformation protocol is available from New England Biolabs [50].

The integrity of all assembled and mutagenized plasmids was verified either by Sanger sequencing (Plateforme de séquençage et de génotypage des génomes, Centre de recherche du centre hospitalier de Québec-Université Laval (CRCHUL), Canada) or by whole plasmid sequencing using Oxford Nanopore Technology with custom analysis and annotation (Plasmidsaurus, USA or Flow Genomics, Canada).

## Plasmid library design

**pRS31N-*ScPDR1* WT:**  *S. cerevisiae PDR1* was expressed from a yeast centromeric plasmid pRS31N-*ScPDR1* that was constructed as follows. First, a plasmid containing the *PDR1* sequence was created, including its promoter and terminator. The plasmid pRS31N containing AMP and NAT resistance markers was doubly digested with 20 U of both HindIII-HF and XbaI restriction enzymes. The insert, a fragment containing *S. cerevisiae PDR1* native promoter (834 bp), coding sequence (3,207 bp), and terminator (345 bp), was amplified from the pMoBY-*ScPDR1* plasmid with primers adding homology to the pRS31N plasmid cloning site (PCR 1). The resulting PCR product corresponding to the insert was incubated for 1h30 at 37 °C with 10 U of DpnI enzyme to remove parental DNA. Subsequently, the plasmid and insert were purified on magnetic beads and assembled by Gibson assembly to obtain pRS31N-*ScPDR1*.

As a BsaI restriction site is used in the construction of the *PDR1* DMS plasmid library, we had to remove three BsaI restriction sites in pRS31N-*ScPDR1*. The three sites were mutated by site-directed mutagenesis (site 1) or by homologous recombination with a short double-stranded DNA fragment (gBlock) corresponding to *PDR1* coding sequence between positions 2238 and 2720 without BsaI restriction sites (sites 2 and 3). The BsaI site 1 was removed by performing site-directed mutagenesis based on the QuickChange Site-Directed Mutagenesis System (Stratagene, La Jolla, CA). We amplified the pRS31N-*ScPDR1* plasmid using a pair of primers containing the desired mutation at the center (PCR 2). The PCR product was then incubated for 1h30 at 37 °C with 10 U of DpnI enzyme to remove parental DNA, and pRS31N-*ScPDR1* mutated plasmids (pRS31N-*ScPDR1* minus site 1) were retrieved directly by transformation in bacteria. For sites 2 and 3, the plasmid pRS31N-Sc*PDR1* minus site 1 was first PCR amplified (PCR 3). The resulting PCR product was incubated for 1h30 at 37 °C with 10 U of DpnI enzyme to remove parental DNA and then, purified on magnetic beads. Finally, the plasmid and gBlock insert were assembled by Gibson assembly [51] to obtain pRS31N-*ScPDR1* WT. For clarity, throughout this study, pRS31N-*ScPDR1* WT refers to the plasmid after the BsaI site replacements, which served as the template for constructing the whole *PDR1* DMS plasmid library.

**oPools:**  The DMS library was generated using only the *PDR1* coding sequence (CDS). Due to the limited length of synthetic DNA and to enable barcode-mutation association with short-read sequencing, the 3,204 bp *PDR1* gene

was divided into 43 fragments of 75 bp each, except the last one which is of 54 bp. Fragment 1 goes from codon 1–25, fragment 2 goes from residue 26–50, and so on, up to the last fragment 43 (the smallest) going from residue 1,050–1,068. For each fragment, a library was built using affordable pools of degenerate oligonucleotides (oPools). oPools are composed of five parts designed as follows (Fig 1A):

1. ~40 bp of homology upstream of the *PDR1* fragment of interest;

2. *PDR1* sequence of the fragment containing a single NNK codon (32 codons, including a stop codon);

3. BsaI module (two BsaI sites separated by CCGAAGCT to avoid self-dimerization);

4. 30 bp DNA barcode with six random N and two fixed sequences (12 bp each) for each codon position. This design facilitates the identification of the position of the mutated codon while allowing multiple distinct barcodes to represent the same mutation. As a result, each unique mutation is linked to several barcodes, providing internal replicates for downstream DMS analyses. oPools currently only allow for nine degenerate positions per DNA fragment. Three are used for the codon and six for the barcode; and

5. the sequencing PBS_i7 site, which is common to all oPools and used for rapid and efficient short-read sequencing library preparation.

To generate the barcodes, we use the following Python script: github.com/lzamparo/DNAbarcodes, which generates all possible 12-nucleotide barcode sequences. From these, 30,000 barcodes with a GC content between 40 and 60% and a Hamming distance of 3 were selected. We then removed barcodes in which BsaI restriction site was present to prevent any problem in the Golden Gate assembly step of our protocol. We also removed barcodes with two identical bases at the beginning or at the end to prevent stretches of the same bases. We generated all individual 30 bp barcodes for the oPools by combining two 12 bp barcodes with six degenerate nucleotides, NN+barcode1+NN+barcode2+NN. We made sure the GC content of the sum of the two barcodes would be ~50%.

All oPool sequences were then generated using a custom Jupyter notebook (opool_generator_20240212.ipynb). We cut the *PDR1* coding sequence into 75 bp fragments (25 codons per fragment), and we identified the 40 bp before and the 4 bp after each fragment. For each fragment, we generated 25 variant oligonucleotides in which a single codon was replaced by an NNK degenerate codon. Each oPool contains 25 different oligonucleotide sequences, each targeting one specific codon. The final sequences were generated by combining the 40 bp before the fragment, the NNK sequence, the 4 bp after the fragment, the BsaI module (GGAGACCgaagCTGGTCTCGACAG), one 30 bp barcode and the PBS_i7 (CTGTCTCTTATACACATCTCCGAGCCCACGAGAC).

All oPools are flanked by homology sequences at both the 5′ end (40 bp upstream of the target fragment within the *PDR1* sequence) and the 3′ end (PBS_i7 sequence, facilitating rapid sequencing library preparation), for a total of 186–258 bp per oPool. The complete list of oPool sequences used in this study is provided in S1 Table.

Because cloning applications require double-stranded DNA inserts, oPools were amplified using two PCR reactions. The first PCR consisted of a single cycle using only the reverse primer (PCR4). Amplicons were purified using magnetic beads at a bead:DNA ratio of 1.8 to enrich for full-length oligonucleotides and to minimize the incorporation of truncated sequences (linked to oPool incomplete synthesis) into the final product. Then, 2 µL of the purified product was used as template for a second PCR, involving both forward and reverse primers and limited to five cycles to reduce PCR-related errors (PCR5). Each oPool amplification was performed in four replicates, which were pooled after the second PCR to maximize the diversity of variants carried into downstream cloning steps. This approach is intended to improve the overall quality of the oPools, ensuring sufficient insert quantity and diversity for subsequent steps of DMS library construction.

## Plasmid library construction

**Cloning strategy of the DMS plasmid library:**  The cloning strategy was divided into two steps: Gibson assembly followed by Golden Gate assembly.

In the first step, the plasmid backbone was amplified from pRS31N-*ScPDR1* WT (PCR 6). The amplified region includes homology arms corresponding to the sequence upstream of the targeted fragment and extends to the plasmid region downstream of the terminator (i7 region). The resulting linear PCR product was incubated for 1h30 at 37 °C with 10 U of DpnI enzyme to remove parental plasmid DNA and then, purified using magnetic beads. The oPools are amplified as described in the oPools section. Backbone and insert fragments were assembled using Gibson assembly protocol. To reach a target of ~25,000 transformants per fragment, necessary to ensure good barcode diversity and mutation coverage (Fig 1), two parallel Gibson reactions (10 μL each) followed by ~5 transformations (3 μL of Gibson reaction per transformation) using NEB 5-alpha Competent *E. coli* cells were performed for each fragment. Following transformations, 5 mL of 2YT medium was added directly onto each agar plate, and colonies were scraped using a sterile glass spreader. All transformants corresponding to a given fragment were pooled together. Aliquots were prepared, each corresponding to an optical density ($OD_{600}$) of 20 per fragment, and stored at −80 °C without (cell pellet) or with (glycerol stock) 15% glycerol. Plasmid libraries were extracted and purified from one cell pellet aliquot per fragment. These purified plasmids served as backbones for the subsequent cloning step.

In the second step, the *PDR1* missing 3′ end region, including the 3′ end coding sequence, terminator, and PBS_i5 (TCGTCGGCAGCGTCAGATGTGTATAAGAGACAG) (Fig 1A), was amplified (PCR 7). BsaI recognition sites were added to both ends of the amplicon to generate non-palindromic overhangs compatible with those in the Gibson-assembled plasmids. The resulting PCR product was incubated for 1h30 at 37 °C with 10 U of DpnI enzyme to remove parental DNA and then, purified on magnetic beads. The previously prepared plasmid libraries served as vectors, and the *PDR1* missing 3′ region acted as inserts for Golden Gate assembly, a one-pot cloning method alternating between BsaI digestion and ligation (PCR 8). To ensure sufficient barcode diversity and adequate mutation coverage (Fig 1), approximately 100,000 transformants per fragment were needed. To achieve this, two parallel Golden Gate reactions (20 μL each) and ~10 transformations (2.5 μL of Golden Gate reaction per transformation) were performed for each fragment using NEB 5-alpha Competent *E. coli* cells, according to the manufacturer's instructions. After transformation, 5 mL of 2YT medium was added directly to each agar plate, and colonies were scraped using a sterile glass spreader. All transformants corresponding to a given fragment were pooled together. Aliquots were prepared, each corresponding to an optical density ($OD_{600}$) of 80 per fragment, and stored at −80 °C without (cell pellet) or with (glycerol stock) 15% glycerol. Plasmid libraries were extracted and purified from one cell pellet aliquot per fragment. The resulting plasmid preparations corresponded to the final *PDR1* DMS plasmid libraries, with a distinct library for each *PDR1* fragment.

**Plasmid library quality control by sequencing:**  At each cloning step, library quality was assessed by high-throughput sequencing, focusing on parameters such as mutation coverage and barcode diversity per mutation (Figs 1C, 1D, and S1–S4).

**Sequencing library preparation:**  To perform sequencing-based quality control after Gibson assembly, we started with 20 ng of plasmid library obtained from bacteria minipreps. Sequencing libraries were prepared via three PCRs. The first two PCRs (PCR9 and PCR10) amplified the fragment of interest along with the downstream DNA barcode, while simultaneously adding the PBS_i5 adapter sequence at the 5′ end. The forward primer included the PBS_i5 and 20 bp upstream of the targeted region, while the reverse primer corresponded only to the PBS_i7. The amplicons were purified, diluted 1:1,000 and used as a template for a third PCR (PCR11), adding the appropriate Illumina i5 and i7 sample indexes for multiplexed sequencing. For each sample, the last PCR was performed in triplicate to maintain the diversity of the sequenced sample. All amplicons from the same sample were pooled and purified using magnetic beads. All samples were sent for high-throughput sequencing (sample description in S5 Table). Although this approach ensures robust amplification of the library and sufficient sequencing depth, it does not directly account for potential amplicon sequencing

biases. Such biases can occur when random DNA molecules are over-amplified compared to others, which can cause significant issues for downstream quantitative analyses. Incorporating Unique Molecular Identifiers (UMIs) during PCR when preparing sequencing libraries would help mitigate the possible occurrence of such biases.

Following Golden Gate assembly, only the DNA barcode region was sequenced, in order to verify that barcode diversity and mutation coverage were maintained. Because the PBS_i5 and PBS_i7 were already present on each side of the DNA barcode, sequencing libraries after Golden Gate assembly were generated in a single PCR (PCR12), which added the appropriate Illumina i5 and i7 sample indexes. Like after Gibson assembly, this PCR was also performed in triplicate per sample to prevent sampling bottlenecks. All replicate amplicons were pooled and purified on magnetic beads prior to sequencing. All samples were then sent for high-throughput sequencing (sample description in S6 Table).

**Sequencing platform:** High-throughput sequencing was performed in paired-end 100 bp or 150 bp at the Centre de recherche du CHU de Québec–Université Laval sequencing platform (CHUL, Canada) using an Illumina NovaSeq 6000 S4 system. Libraries were sequenced at an average depth of ~250 reads per expected variant to assess barcode diversity and mutation coverage.

**Sequencing read processing:** To analyze quality control after Gibson assembly, first, low-quality reads were removed with *fastp* using default parameters -q 15 (the minimum phred score quality value that a base is qualified equal to 15) and -u 40 (40% of bases allowed to be unqualified). Reads were then merged with *bbmerge* from *bbmap*. All subsequent processing was done using custom Python scripts available at https://github.com/Landrylab/Jann_et_al_2025. To remove potential sequencing errors, the following reads were discarded: reads with incorrect length (reads including insertions or deletions), reads with more than one mutated codon within the *PDR1* coding region (75 bp), reads with unexpected mutations within barcode sequences, and reads with missing nucleotides within the *PDR1* or barcode sequence. After filtering, more than 60% and 80% of all reads were retained for F13 and F43, respectively.

Mutations with coverage of less than two reads were discarded. To measure the impact of combining results from multiple transformations (16 reactions, each with 5,000 (F13) or 7,500 (F43) transformants), we calculated the cumulative % of different metrics across all transformations sorted in a randomized manner. Means and confidence intervals were obtained for 100 random draws of transformation reactions. Metrics included: (i) informative barcodes, (ii) mutation coverage, (iii) barcode diversity: #barcodes >4 per mutation, and (iv) barcode diversity: #barcodes >9 per mutation. To standardize data, all metrics have been converted to the number of transformants/bp.

As done following the Gibson assembly, raw reads obtained after Golden Gate assembly product sequencing were filtered using *fastp* (default parameters), and overlapping paired-end reads were merged using *bbmerge.* Reads with incorrect length, unexpected barcode sequences, or missing nucleotides were removed, along with single-count reads. Barcode-mutation associations were performed by merging barcode reads from the Golden Gate assembly with the reference association generated after the Gibson assembly. To evaluate the library quality after the Golden Gate step, we re-calculated key metrics: informative barcodes, mutation coverage, and barcode diversity.

## Assessment of plasmid library quality, diversity, and uniformity

To evaluate the quality and representation of variants in the constructed libraries, we quantified their uniformity using two established metrics. First, the Gini coefficient, an index of distribution bias ranging from 0 (perfect uniformity) to 1 (maximum bias) [31], was calculated to assess the equal representation of variants after each cloning step (Gibson and Golden Gate). Second, we used the uniformity score proposed by [28], defined as the logarithmic difference between the 90th and 10th percentiles of the read count for each variant per fragment, where a small value indicates greater uniformity. These analyses were performed on the plasmid sequencing data to confirm a balanced distribution of variants before functional selection (S10 and S11 Figs; S7 Table).

The presence of all expected mutations, with excellent DNA barcode diversity per mutation was confirmed in libraries F13 and F43 (after Gibson assembly: all mutations are covered by #barcodes >4 and after Golden Gate

assembly:~93.6%–100% of mutations are covered by #barcodes >4) (Figs 1C, 1D, S1, and S2). A small number of mutations, specifically those at residues 322–324, were detected at lower read counts. Nevertheless, they still showed high mutation coverage (98.3%) and acceptable barcode diversity, with 71.7% of mutations with #barcodes >4 (Figs 1D and S2), which is the threshold required to have over 0.9 in terms of correlation coefficient between the selection coefficients measured from F13 directly and the barcodes: (Fig 2C).

Although the main text presents in detail only the characterization of the mutant libraries for two representative *PDR1* fragments (F13 and F43), we also carried out, in parallel, the construction of a comprehensive plasmid mutant library covering the entire *PDR1* coding sequence (total of 43 libraries, one for each fragment of 25 aa). This extended library was generated using the same two-step cloning strategy described above.

Analysis of the complete 43-fragment library confirmed strong performance across all quality metrics (S5–S8 Figs; S8 Table). Nearly all expected mutations (~98,4%) were recovered, with only a few absent positions likely attributable to synthesis dropout in the oPools. Barcode diversity per mutation remained high throughout all libraries: after Gibson assembly,~97,5% mutations were represented by #barcodes >4, and after Golden Gate assembly this remained true for ~95.4% of positions (S5–S8 Figs; S8 Table). Uniformity analysis further indicated that variant abundance across the library (Gini coefficient ~0.32) was well balanced and acceptable for downstream DMS experiments (S8 Table).

Consistent with previous observations using NNK degenerate codon libraries, expected biases in variant diversity were observed across amino acid substitutions as some amino acids are encoded by more codons. Amino acids such as leucine, arginine, and serine are encoded by three codons, while most others are encoded by one or two, resulting in higher expected representation for these residues (Figs 1C, 1D, S5, and S6).

However, codon-level analyses revealed that synonymous codon multiplicity alone does not fully explain the observed patterns (S3, S4, S7, and S8 Figs). Certain codons rich in thymine or guanine (e.g., TTT, GGG, TGG, TTG) consistently showed higher barcode diversity, while others were underrepresented, such as the codons for proline, threonine, and alanine (S3–S8 Figs). In addition, position-specific biases were detected along the gene, with a small number of codon positions showing reduced (*PDR1* F13–codons 301 and 322–324: S3 and S4 Figs) or absent representation (19 codon positions in the entire *PDR1* gene: S5–S8 Figs), likely reflecting dropouts during oPool synthesis rather than cloning inefficiencies.

We additionally quantified the representation of non-programmed variants and found that ~10% of reads corresponded to the WT sequence and ~10% to double-mutants (S8 Table). These species can either be excluded during downstream fitness estimation or leveraged for dedicated analyses of epistatic interactions, depending on the experimental aims.

To support reproducibility and provide a broader overview of the experimental workflow, the results and quality metrics for each cloning stage of the full-length *PDR1* library are presented in S8 Table and S5–S8 Figs. In addition, we provide a detailed estimated experimental timeline for the construction of the full 43-fragment *PDR1* mutant library in S9 Table. This resource is intended to serve as a practical reference for researchers aiming to apply or adapt such an approach in future DMS projects.

**Quality control of final plasmid libraries:** To assess the accuracy of the cloning strategy and evaluate potential PCR-induced mutations, we performed Whole Plasmid Sequencing of six plasmids derived from independent fragments assembly by Golden Gate. No unintended mutations were present in the *PDR1* promoter, coding sequence, or terminator regions. This quality-control step validates the reliability of our cloning approach. Although pre-sequencing of all PCR-amplified fragments prior to Golden Gate assembly could further ensure accuracy, it was not performed here due to cost and scalability considerations.

## Variant library construction

Even if *PDR1* is not essential for *S. cerevisiae* viability, we replaced its endogenous promoter with a DOX-repressible promoter (tetO7). This allowed inhibition of genomic *PDR1* expression and complementation by plasmid-encoded variant

libraries. We chose this approach after observing low transformation efficiency when attempting to transform a yeast strain harboring a complete *PDR1* deletion.

**Yeast strain construction (S12A Fig):** The *ScPDR1*-DOX strain was generated based on the Yeast Tet-Promoters Hughes Collection (yTHC) [52]. The endogenous promoter of *PDR1* (spanning 50 bp upstream of the ATG codon; chrVIII:121683-121733) was replaced with a tetracycline-regulatable promoter. The KANMX-tetO7 cassette was amplified from plasmid pKB33 (PCR13) and introduced into competent *S. cerevisiae* R1158 cells using an adapted lithium acetate transformation protocol [53]. Transformants were selected on YPD medium containing 200 µg/mL G418.

**Validation of genomic *PDR1* expression control in *ScPDR1*-DOX:** To validate the control of genomic *PDR1* expression in the *ScPDR1*-DOX strain, *PDR1* expression levels were quantified in the presence and absence of DOX in the culture medium (S12B Fig). For this purpose, the gene coding for a monomeric enhanced Green Fluorescent Protein (*mEGFP*) was fused to the *PDR1* coding sequence in the genome of *ScPDR1*-DOX strain. The *mEGFP* sequence along with a hygromycin resistance module was amplified from plasmid pFA-*mEGFP-HPHNT1* (see construction below) using primers introducing homology arms targeting the 3′ end of the *PDR1* CDS and its terminator (PCR 14). Amplicon was treated with 20 U of DpnI for 1 h at 37 °C, purified using magnetic beads, and used to transform *ScPDR1*-DOX competent cells using the standard lithium acetate transformation protocol [53].

Yeast strains (*ScPDR1-GFP*-DOX and *ScPDR1*-DOX) were cultured overnight at 30 °C with shaking in synthetic complete (SC) medium in 24 deep-well plates. The SC medium contained 0.17% yeast nitrogen base without amino acids and ammonium sulfate, 0.1% monosodium glutamate, 2% glucose, and amino acids drop-out without tryptophan. Saturated cultures were diluted to an $OD_{600}$ of 0.15 in fresh medium with or without DOX (10 µg/mL), and grown again under agitation at 30 °C until reaching an $OD_{600}$ of 0.5. Cells were then diluted to ~500 cells/µL in 0.2 µm filtered distilled water for flow cytometry analysis. GFP fluorescence was measured using a Guava easyCyte flow cytometer (MilliporeSigma) with excitation at 642 nm (GRN-B channel, 525/30 filter). A preliminary gating step (FSC-H (Forward Scatter Height) <15,000 and FSC-A (Forward Scatter Area) <15,000) was applied to select the main population of morphologically normal cells and to exclude non-representative events. A total of 5,000 events were recorded per replicate. Three independent biological replicates were analyzed per condition. Fluorescence signals were processed using a custom Python script and all the data (including control lacking GFP) are shown in S12B Fig.

In order to generate *ScPDR1-GFP*-DOX yeast strain, we first had to construct pFA-*mEGFP-HPHNT1* plasmid. This was done by inserting *mEGFP* gene (insert 1) and *ENO1* terminator fragment (insert 2) into pFA-*hphNT1* (vector) using Gibson assembly. All three fragments were PCR amplified (PCR15, PCR16, and PCR17, respectively) and treated with 20 units of DpnI for 1 h at 37 °C before being purified on magnetic beads.

**Yeast strain validation by complementation:** In order to validate whether *PDR1* expressed from the plasmid complements its genomic counterpart, we selected a condition where *PDR1* expression is beneficial (in presence of antifungal) and assessed the growth of *ScPDR1*-DOX strain transformed either with an empty plasmid or with one containing *PDR1* WT sequence (S12C Fig). After adjusting cell density of precultures to an $OD_{600}$ of 1, three serial 1/5 dilutions were prepared in 200 µL of water (i.e., 40 µL of cells in 160 µL of water). Then, 5 µL of each dilution were spotted on YPD + NAT agar plates supplemented with either 0.1% DMSO (control), 1 µg/mL ITR, 2 µg/mL ITR, or 4 µg/mL ITR, in the presence or not of DOX (10 µg/mL). Plates were incubated at 30 °C for 48 h before imaging.

**Library in yeast:** The DMS plasmid library was introduced into *ScPDR1*-DOX yeast cells via lithium acetate transformation, performed fragment by fragment using a standard protocol [53]. For each fragment, approximately 100,000 colonies were recovered to ensure sufficient library coverage and barcode diversity. To collect transformants, 5 mL of YPD medium was added directly to each plate, and colonies were scraped using a sterile glass spreader. All transformants corresponding to a given fragment were pooled together. Aliquots were prepared, each corresponding to an optical density ($OD_{600}$) of 80 per fragment, and stored at −80 °C in 25% glycerol.

## Pooled competition assay

A pooled competition assay was performed using the *PDR1* F13 variant library, in the presence or absence of an antifungal. All cultures were grown in YPD medium supplemented with NAT (100 µg/mL), to maintain plasmid selection, and DOX (10 µg/mL), to repress the endogenous *PDR1* promoter. As the used antifungal compound was dissolved in DMSO, the control condition also included 0.1% DMSO. POSA was used at a final concentration of 0.40 µg/mL, corresponding to the concentration that inhibits approximately 50% of WT growth, as previously reported [5].

The *PDR1* F13 variant library was first pre-cultured overnight in YPD + NAT, starting with an estimated inoculum of ~10,000 cells per variant (Time Point 0, TP0). Saturated cultures were diluted to an initial $OD_{600}$ of 0.05 into fresh YPD + NAT + DOX + POSA, and grown until an $OD_{600}$ of 0.80 was reached (TP1). A second dilution and growth cycle under the same conditions was performed ($OD_{600}$ = 0.05 to 0.80), corresponding to a total of approximately eight mitotic generations (TP2). Identical conditions were used for the control (DMSO) condition.

At TP0, TP1, and TP2, cells were harvested by collecting 5 $OD_{600}$ units per sample in 24 deep-well plate, followed by centrifugation at 916*g* for 5 min. Pellets were resuspended in 1 mL of YPD, transferred to 1.5 mL microcentrifuge tubes, and centrifuged again at 600*g* for 2 min. Supernatants were removed by aspiration, and resulting pellets were stored at −80 °C until plasmid extraction.

## Plasmid recovery

Plasmid recovery from yeast was performed using the Zymoprep Yeast Plasmid Miniprep II kit, with the following modified protocol based on [5,24] to improve yield. Frozen cell pellets were resuspended in 200 µL of Solution 1, followed by the addition of 30 U of Zymolyase 20T. Cells were gently vortexed and incubated at 37 °C for 2 h. Following incubation, tubes were briefly vortexed and frozen at −80 °C for at least 20 min (up to overnight). After thawing at 37 °C for 3 min, 200 µL of Solution 2 was added and mixed, followed by the addition of 400 µL of Solution 3. Tubes were centrifuged at maximum speed for 3 min, and the supernatant (~800 µL) was transferred to a Zymo-Spin I column. Columns were washed with 550 µL of ethanol-containing wash buffer and centrifuged for 1.5 min at maximum speed. After removing the flow-through, columns were spun for an additional min to dry. Plasmids were eluted by adding 10 µL of Tris buffer (10 mM, pH 8.0) and centrifuging for 1 min at maximum speed. Eluted DNA was stored at −20 °C until further processing.

**High-throughput sequencing:** To validate whether DNA-barcode inference mutations corresponded accurately to the actual mutations present in the F13 sequence of *PDR1*, two types of library were generated: one containing the F13 region of *PDR1* and one consisting of the barcodes.

Sequencing from yeast minipreps was performed using a protocol similar to that described in the Plasmid library quality control by sequencing section. Starting from 3 µL of yeast miniprep, an initial PCR was carried out to amplify only *PDR1* from the plasmid, using Phusion High-Fidelity DNA Polymerase and M13 primers flanking the *PDR1* insert on pRS31N (PCR18). This step avoids sequencing contaminating genomic *PDR1* and ensures that the same sequencing material could be used for both F13 region and barcode-based analyses, allowing a direct and unbiased comparison between the two datasets. In a standard barcode-based DMS experiment, the DNA-barcodes could be directly amplified using the PBS_i5 and PBS_i7 flaking sites, which would simplify library preparation and improve sequencing data quality.

To generate the F13 sequencing libraries, two rounds of PCR were then performed. Starting with 2 µL of M13-purified amplicons, a first PCR made with Phusion High-Fidelity DNA Polymerase (PCR19) was used to amplify the F13 region while introducing PBS_i5 and PBS_i7 sequences at the 5′ and 3′ ends, respectively. In addition, a 0–3 nucleotides spacer (N) was added to the primers to increase sequencing diversity. A second PCR (PCR20) was then used to add the appropriate Illumina i5 and i7 sample indexes, as described in the Plasmid library quality control by sequencing section.

To generate the DNA barcode sequencing libraries, 2 μL of the M13-purified amplicons were used as a template for a single PCR reaction to add the appropriate Illumina i5 and i7 indexes (PCR21), as described in the Plasmid library quality control by sequencing section.

Both the F13 and DNA barcode sequencing libraries were sent for high-throughput sequencing in paired-end 150 bp at the Centre de recherche du CHU de Québec - Université Laval sequencing platform (CHUL, Canada) using an Illumina NovaSeq 6000 S4 flow cell (sample description in S10 Table). All samples were sequenced to achieve a minimum of 500 reads per expected variant (i.e., ~0.4 million reads per sample for the F13 libraries and ~4 million reads per sample for the DNA barcode libraries). Only samples from timepoints TP0 and TP2 were sequenced. TP1 was excluded, as the limited frequency changes observed after just four generations hindered robust genotype-phenotype associations.

### Data analysis

**Selection coefficient and variant characterization:** Selection coefficients were obtained with *gyōza*, a Snakemake-based workflow for the analysis of deep mutational scanning (DMS) data [54]. The pipeline includes adapter trimming with *Cutadapt*, read merging using *PANDAseq*, and clustering of identical reads via *VSEARCH*. Singleton reads (i.e., sequences observed only once) were excluded to reduce noise from sequencing errors. A custom script then aggregated read statistics across all processing steps.

The F13 samples were processed using the 'codon' mode of gyōza v1.1.8 [54], with constant 20 bp sequences on either side of the mutated locus specified for trimming. For the barcoded library, the "barcode" mode of *gyōza* v1.1.8 was used instead, by providing the dataframe of barcode-variant associations. Briefly, (i) all singletons (read count of 1) were discarded, (ii) $\log_2$ fold changes of allele frequencies were calculated between TP0 and TP2, (iii) selection coefficients were calculated by normalizing with the number of mitotic generations (to account for different growth rates across cultures and make sure functional impact scores reflect the per-generation selective advantage), and (iv) selection coefficients were further normalized by subtracting the median $\log_2$ fold-change of silent variants (excluding the WT nucleotide sequence).

Variants were assigned confidence scores (as defined by *gyōza*) based on their initial read count at TP0 across replicates. A minimum threshold of five reads per nucleotide sequence was used to define high-confidence variants, which were retained for downstream analysis. Selection coefficients were then aggregated by amino acid position, averaging across synonymous codons (S5 and S6 Data). The resulting tables were used to classify mutations from resistant to sensitive (0.6 to −0.6) based on their deviation from the WT. Finally, heatmaps were generated using the selection coefficient for each variant to enable clear visualization and comparison between F13 region and DNA barcode sequencing (Figs 2B and S14).

To assess the quality of the barcoded library, we compared the final selection coefficients per mutation obtained from sequencing of either mutated F13 region or DNA-barcode inference. To evaluate how barcode diversity influences the reliability of these coefficients, we progressively increased the number of barcodes required per mutation (from 1 to 10) and, at each level, performed 100 random samplings. To ensure consistent comparisons across variants with different barcode counts, we implemented a standardized subsampling procedure during correlation analysis. If a variant had fewer barcodes than the target number (*n*), some barcodes were randomly resampled multiple times to reach *n*. Conversely, if a variant had more than *n* barcodes, a random subset of *n* barcodes was selected. This approach allowed us to normalize barcode representation across variants and to accurately assess correlation metrics based on 100 independent subsamplings. For each sampling, we calculated the correlation between selection coefficients inferred from DNA barcodes and those obtained from the F13 region (Fig 2C). An analysis of the correlation stratified by the number of barcodes per variant confirms the robustness of our strategy, even for variants associated with a lower barcode count (S14 Fig).

As no experimentally determined structure was available for Pdr1 in the Protein Data Bank (PDB, RCSB), we used the AlphaFold 3 server [55] to predict its tertiary structure (pTM score = 0.72; seed = 264,428,788) Fig 2D The amino acid

sequence of *S. cerevisiae* Pdr1 was obtained from UniProt (accession: P12383). The predicted structure was visualized using ChimeraX [56].

## Supporting information

**S1 Fig. Barcode diversity per amino acid substitution after Gibson assembly.** Heatmaps show the total number of unique barcodes for each possible amino acid substitution at each codon position in Pdr1 fragments F13 and F43. For each fragment, a total of 25,000 transformants were recovered and analyzed. Covered mutations are mutations represented by #barcodes >0, while barcode diversity represented by #barcodes >4 or >9 per mutation demonstrates an increasing number of replicates for a same mutation. Gray squares represent WT amino acids. In the heatmaps, the number of barcodes per mutation is clipped at 10. The numerical data underlying this graph is provided in S2 Data. (TIF)

**S2 Fig. Barcode diversity per amino acid substitution after Golden Gate assembly.** Heatmaps show the total number of unique barcodes for each possible amino acid substitution at each codon position in Pdr1 fragments F13 and F43. For each fragment, a total of 100,000 transformants were recovered and analyzed. Covered mutations are mutations represented by #barcodes >0, while barcode diversity represented by #barcodes >4 or >9 per mutation demonstrates an increasing number of replicates for a same mutation. Gray squares represent WT amino acids. In the heatmaps, the number of barcodes per mutation is clipped at 10. The numerical data underlying this graph is provided in S3 Data. (TIF)

**S3 Fig. Barcode diversity per NNK codon substitution after Gibson assembly.** Heatmaps show barcode diversity for each possible NNK codon substitution at each codon position in Pdr1 fragments F13 and F43. For each fragment, a total of 25,000 transformants were recovered and analyzed. Barcode diversity is shown using an unclipped color scale, allowing visualization of the full range of barcode counts. Mutations covered by high barcode diversity (#barcodes ≥10 and ≥4) are represented by a blue and purple scale, respectively, while lower barcode diversity (#barcodes <4) is represented by a red scale. Gray squares represent WT amino acids. The numerical data underlying this graph is provided in S2 Data. (TIF)

**S4 Fig. Barcode diversity per NNK codon substitution after Golden Gate assembly.** Heatmaps show barcode diversity for each possible NNK codon substitution at each codon position in Pdr1 fragments F13 and F43. For each fragment, a total of 100,000 transformants were recovered and analyzed. Barcode diversity is shown using an unclipped color scale, allowing visualization of the full range of barcode counts. Mutations covered by high barcode diversity (#barcodes ≥10 and ≥4) are represented by a blue and purple scale, respectively, while lower barcode diversity (#barcodes <4) is represented by a red scale. Gray squares represent WT amino acids. The numerical data underlying this graph is provided in S3 Data. (TIF)

**S5 Fig. Library quality control per amino acid substitution after Gibson assembly of the full-length *PDR1* sequence (43 fragments).** Heatmap displays the barcode diversity associated with each possible amino acid substitution at each codon position. Barcode diversity is shown using an unclipped color scale, allowing visualization of the full range of barcode counts. Mutations covered by high barcode diversity (#barcodes ≥10 and ≥4) are represented by a blue and purple scale, respectively, while lower barcode diversity (#barcodes <4) is represented by a red scale. Gray squares represent WT amino acids. The numerical data underlying this graph is provided in S2 Data. (PNG)

**S6 Fig. Library quality control per amino acid substitution after Golden Gate assembly of the full-length *PDR1* sequence (43 fragments).** Heatmap displays the barcode diversity associated with each possible amino acid substitution

at each codon position. Barcode diversity is shown using an unclipped color scale, allowing visualization of the full range of barcode counts, including low values in red. Mutations covered by high barcode diversity (#barcodes ≥10 and ≥4) are represented by a blue and purple scale, respectively, while lower barcode diversity (#barcodes <4) is represented by a red scale. Gray squares represent WT amino acids. The numerical data underlying this graph is provided in S3 Data.
(PNG)

**S7 Fig. Library quality control per NNK codon substitution after Gibson assembly of the full-length *PDR1* sequence (43 fragments).** Heatmaps show barcode diversity for each possible NNK codon substitution at each codon position. Barcode diversity is shown using an unclipped color scale, allowing visualization of the full range of barcode counts. Mutations covered by high barcode diversity (#barcodes ≥10 and ≥4) are represented by a blue and purple scale, respectively, while lower barcode diversity (#barcodes <4) is represented by a red scale. Gray squares represent WT amino acids. The numerical data underlying this graph is provided in S2 Data.
(PNG)

**S8 Fig. Library quality control per NNK codon substitution after Golden Gate assembly of the full-length *PDR1* sequence (43 fragments).** Heatmaps show barcode diversity for each possible NNK codon substitution at each codon position. Barcode diversity is shown using an unclipped color scale, allowing visualization of the full range of barcode counts, including low values in red. Mutations covered by high barcode diversity (#barcodes ≥10 and ≥4) are represented by a blue and purple scale, respectively, while lower barcode diversity (#barcodes <4) is represented by a red scale. Gray squares represent WT amino acids. The numerical data underlying this graph is provided in S3 Data.
(PNG)

**S9 Fig. Distribution of barcode counts per amino-acid mutation after (A) Gibson and (B) Golden Gate assemblies for Pdr1 F13 and F43.** Histograms show the number of unique barcodes associated with each amino-acid substitution in fragments F13 (gray) and F43 (blue). For each fragment, the x-axis indicates the number of barcodes linked to a given amino-acid substitution, and the y-axis shows the number of substitutions observed at each barcode count. The numerical data underlying these graphs is provided in S2 and S3 Data.
(TIF)

**S10 Fig. Assessment of library uniformity after each cloning step (Gibson and Golden Gate assemblies) using the Gini coefficient.** Boxplots represent the distribution of Gini coefficients (ranging from 0 to 1) calculated for each fragment, where lower values indicate more uniform coverage and higher values indicate increased inequality in representation. The numerical data underlying this graph is provided in S4 Data.
(TIF)

**S11 Fig. Assessment of library uniformity after each cloning step (Gibson and Golden Gate assemblies) using the uniformity score.** Boxplots show the distribution of uniformity scores for each fragment, defined as the log difference between the 90th and 10th percentiles of mutant read counts. Lower scores indicate more uniform representation, while higher scores indicate greater inequality. The numerical data underlying this graph is provided in S4 Data.
(TIF)

**S12 Fig. Construction and validation of a *S. cerevisiae* strain to measure the impact of *PDR1* mutations on drug resistance. A) Inducible control of genomic *PDR1* expression in *Saccharomyces cerevisiae* using a doxycycline-repressible promoter (TetO7).** Created in BioRender. Barff, T. (2025) https://BioRender.com/zo8wlgo. **B) Validation of genomic *PDR1* repression in the *ScPDR1*-DOX strain.** *PDR1* expression level was assessed by flow cytometry using a *GFP* fusion as a reporter. Fluorescence intensity is shown for the strain expressing GFP (left panel: *ScPDR1-GFP*-DOX) and the negative control strain without *GFP* (right panel: *ScPDR1*-DOX) in media with (red) or

without (blue) DOX. GFP intensity was analyzed on the main population of morphologically normal cells (FSC-H<15000 and FSC-A<15000). 5,000 events were recorded per replicate. Three independent biological replicates were analyzed per condition. The numerical data underlying this graph is provided in S8 Data. **C)** in *ScPDR1*-DOX by plasmid-expression of *ScPDR1* (pRS31N-*ScPDR1* WT) along with the control strain (pRS31N-empty). Growth conditions: DMSO (control) or Itraconazole (ITR) (antifungal).
(TIF)

**S13 Fig. Correlation between the selection coefficients measured with direct sequencing of the mutated *PDR1* F13 coding region and DNA-barcode inference, using all usable barcodes.** The strong correlation (Pearson $r = 0.97$) indicates that barcode-based estimates accurately recapitulate the directly measured fitness effects. The number of barcodes per mutation does not affect the correlation. The numerical data underlying this graph is provided in S5 and S6 Data.
(TIF)

**S14 Fig. Correlation between the selection coefficients measured with direct sequencing of the mutated *PDR1* F13 coding region and DNA-barcode inference, as a function of the number of barcodes per variant considered.** Curves correspond to variants with exactly 5 ($n = 40$), 10 ($n = 25$), or 15 ($n = 17$) barcodes. Means and confidence intervals were calculated from 100 random barcodes subsamplings per variant. The high correlation observed even for variants with only five barcodes indicates that the overall correlation is not driven only by highly diversified variants. The numerical data underlying this graph is provided in S9 Data.
(TIF)

**S1 Table. oPool design.**
(XLSX)

**S2 Table. Key resources.**
(XLSX)

**S3 Table. Oligonucleotide sequences.**
(XLSX)

**S4 Table. PCR protocols.**
(XLSX)

**S5 Table. Gibson sequencing sample description.**
(XLSX)

**S6 Table. Golden Gate sequencing sample description.**
(XLSX)

**S7 Table. Comparison of library uniformity (TP0) between two DMS plasmid library construction strategies.** This table presents a quantitative comparison of the variant uniformity achieved by our scalable barcoded strategy *PDR1* DMS TP0 F13 library and a reference DMS plasmid library *CaERG11* previously generated in our laboratory [5]. The *CaERG11* library was constructed using a highly precise, low-throughput method: DNA fragments synthesized by Twist Bioscience were cloned codon-by-codon into a yeast expression vector, and the individual codon libraries were then pooled together proportionally based on the number of mutants. This meticulous process allows to achieve a near-perfect uniformity of variant representation, which serves here as a gold-standard reference. The comparison indicates that the uniformity achieved by our barcoded strategy is certainly less efficient than the *CaERG11* reference, but it is perfectly acceptable for robust DMS experiments (Gini coefficient around 0.52). Crucially, the data demonstrate that our method achieves this

acceptable uniformity while remaining substantially more cost-effective and scalable. The metrics computed include the Gini coefficient ($0 =$ perfect uniformity, $1 =$ maximum bias) and the Uniformity Score.
(XLSX)

**S8 Table. Summary of coverage statistics for the two cloning steps for all libraries of the complete *PDR1* gene.**
(XLSX)

**S9 Table. Estimated experimental timeline to construct the complete plasmid mutant *PDR1* library.**
(XLSX)

**S10 Table. DMS sample description.**
(XLSX)

**S1 Data. Barcode diversity and mutation coverage analysis per transformants.**
(XLSX)

**S2 Data. Barcode diversity per after Gibson assembly.**
(XLSX)

**S3 Data. Barcode diversity per after Golden Gate assembly.**
(XLSX)

**S4 Data. Uniformity scores after Gibson and Golden Gate assemblies.**
(XLSX)

**S5 Data. *Gyoza* all scores based on *PDR1* F13 region direct sequencing.**
(XLSX)

**S6 Data. *Gyoza* all scores based on DNA barcodes sequencing.**
(XLSX)

**S7 Data. Correlation of selection coefficients between *PDR1* F13 direct sequencing and DNA barcodes sequencing from barcode subsampling.**
(XLSX)

**S8 Data. Individual flow cytometry event data used to generate the *PDR1*-DOX and *PDR1*-GFP-DOX graphs.** This table contains FSC, SSC, and GFP fluorescence measurements, including log10-transformed GFP values, for all events contributing to the summarized distributions shown in S12 Fig.
(XLSX)

**S9 Data. Correlation of selection coefficients between *PDR1* F13 direct sequencing and DNA barcodes sequencing from barcode subsampling for variants with 5, 10, or 15 barcodes.**
(XLSX)

## Acknowledgments

We thank Dan Evans-Yamamoto, Philippe Després, and other Landrylab members for discussions and feedback on the design of the cloning strategy.

## Author contributions

**Conceptualization:** Jessica Jann, Isabelle Gagnon-Arsenault, Alexandre K. Dubé, Christian R. Landry.

**Data curation:** Jessica Jann, Isabelle Gagnon-Arsenault.

**Formal analysis:** Jessica Jann, Alicia Pageau, Anna Fijarczyk, Romain Durand.

**Funding acquisition:** Christian R. Landry.

**Investigation:** Jessica Jann, Isabelle Gagnon-Arsenault, Christian R. Landry.

**Methodology:** Jessica Jann, Isabelle Gagnon-Arsenault, Alexandre K. Dubé.

**Project administration:** Christian R. Landry.

**Resources:** Jessica Jann.

**Supervision:** Christian R. Landry.

**Validation:** Jessica Jann.

**Visualization:** Jessica Jann, Alicia Pageau.

**Writing – original draft:** Jessica Jann, Isabelle Gagnon-Arsenault.

**Writing – review & editing:** Jessica Jann, Isabelle Gagnon-Arsenault, Christian R. Landry.

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
