## [Editor Report · Decision Letter 0]

20 Jun 2025

Dear Dr Landry,

Thank you for submitting your manuscript entitled "Making deep mutational scanning accessible: a cost-efficient approach to construct barcoded libraries for genes of any length" for consideration as a Methods and Resources article by PLOS Biology. Please accept my apologies for the delay in getting back to you, as we consulted with an academic editor about your submission.

Your manuscript has now been evaluated by the PLOS Biology editorial staff, as well as by an academic editor with relevant expertise, and I am writing to let you know that we would like to send your submission out for external peer review.

Once your full submission is complete, your paper will undergo a series of checks in preparation for peer review. After your manuscript has passed the checks it will be sent out for review. To provide the metadata for your submission, please Login to Editorial Manager (https://www.editorialmanager.com/pbiology) within two working days, i.e. by Jun 22 2025 11:59PM.

Kind regards,

Richard

Richard Hodge, PhD

rhodge@plos.org

PLOS

---

## [Decision Letter · Decision Letter 1]

24 Jul 2025

Dear Dr Landry,

Thank you for your patience while your manuscript "Making deep mutational scanning accessible: a cost-efficient approach to construct barcoded libraries for genes of any length" was peer-reviewed at PLOS Biology as a Methods and Resources Article. Please accept my sincere apologies for the delays that you have experienced during the peer review process. Your manuscript has now been evaluated by the PLOS Biology editors, an Academic Editor with relevant expertise, and by four independent reviewers.

In light of the reviews, which you will find at the end of this email, we would like to invite you to revise the work to thoroughly address the reviewers' reports.

As you will see, the reviewers are generally positive about your library construction approach and think it will be useful for the DMS field. Reviewer’s #1 and #3 raise some overlapping concerns with the claim that the method can be applied to whole genes and Reviewer #3 asks that a direct demonstration of its application to sizable sequence length is provided, as well as quantifying variant representation biases. In addition, Reviewer #1 notes that a more thorough evaluation and comparison to the current literature should be provided to contextualize the methodological advance. Finally, Reviewer’s #2 and #4 are very positive and provide a few minor comments to strengthen the reporting and presentation.

Given the extent of revision needed, we cannot make a decision about publication until we have seen the revised manuscript and your response to the reviewers' comments. Your revised manuscript is likely to be sent for further evaluation by all or a subset of the reviewers.

**IMPORTANT - SUBMITTING YOUR REVISION**

*Re-submission Checklist*

*Published Peer Review*

*PLOS Data Policy*

*Blot and Gel Data Policy*

Best regards,

Richard

Richard Hodge, PhD

rhodge@plos.org

REVIEWS:

Reviewer #1: Jann et al. present a method for constructing saturation mutagenesis libraries coupled to nucleic acid barcodes. They demonstrate the efficacy of this method by performing construction of two barcoded NNK tiles (one 75-nt, one 54-nt - (Tile 43, Supporting Tables)), and performing a deep mutational scanning experiment on one 75-nt tile of the PDR1 sequence. This motivation behind this method is the need for cost-effective deep mutational experiments of full-length genes (for example, the coding sequence for PDR1 is 3204 bp). Two major strategies exist - one can either mutate each tile separately and perform separate selections and then sequencing ('tile-seq'). Alternatively, one can couple a nucleic acid barcode to a specific variant of interest and sequence the barcode (for an early example see Sarkisyan, Karen S., et al. "Local fitness landscape of the green fluorescent protein." Nature 533.7603 (2016): 397-401.). The Sarkisyan paper used a molecular biology 'hack' to connect the barcode to a variant using short read sequencing; there are other - conceptually similar - short read coupling strategies as well (Petersen …Whitehead Nat Comms 2024). Alternatively, there are methods to pre-assign a barcode to a gene variant, avoiding barcode-variant haplotyping altogether (Ranganathan Nat Comms 2019). More commonly, the barcode is coupled to the variant of interest using long read sequencing by PacBio or Oxford Nanopore. The disadvantage of the long read strategy is the sequencing cost and throughput. In the author's strategy, they use a two-step method where a barcode is covalently fused to a 75-nt gene tile. In step 1, a Gibson assembly is performed. In step 2, the full plasmid is reconstituted via Golden Gate assembly using the 3' portion of the remaining gene. The advantage of the Jann method is resolving the barcode to the gene variant after step 1 via short read sequencing. The major disadvantage of their method is in the need for n separate long-length PCRs to generate the linearized DNA for Gibson assembly in step 1 (here n is the number of tiles). The paper flows well and has well-described and executed experiments. However - as described below - it is unclear on reading whether this method represents an improvement over current methods for scanning long full-length genes. A more thorough evaluation and comparison to the current literature, as well as additional analysis would improve the quality of this contribution.

Major concerns

1. The method is specifically designed to address mutagenesis of very large genes. However, the paper does not appreciably contextualizing similarities/differences between this method and previous methods, and the introduction does not place the present work with the current literature on technical improvements for large genes. For example, it appears that the method presented here is very similar to the method employed by Sri Kosuri et al in two papers, one of which is cited by the authors of the present manuscript: [Jones et al., 2020, eLife] and [Howard et al., 2025, eLife]. The authors should specify if indeed the method is the same, or if not, how it differs. What are the distinguishing features? What are the strengths of this short read haplotyping strategy vs. other approaches that exist in the literature? Apt comparisons would be with nicking mutagenesis (Mighell et al., 2023, PLoS One) or DIMPLE (Macdonald et al., 2023, Genome Biology). These methods are much more widely used, can incorporate NNK encoded nucleotides, and are alternatives for generating the library when coupled with barcodes and long read haplotyping.

2. The authors list 3 parameters key for satmut libraries: informative barcodes, mutation coverage, and barcode diversity. However, one parameter is missing from this list: uniformity of variants. This is important for limiting sequencing costs: two libraries with the same percent of informative barcodes, mutation coverage, and barcode diversity could have very different uniformity, leading to more sequencing reads required to assess the less uniform one. Metrics exist for evaluating uniformity in tiled gene libraries (e.g., Mighell et al., 2023, PLoS One), and would encourage an analysis of tile 13 and 43 libraries using this or similar metrics.

3. My suggestion is to remove the cost comparison or to greatly strengthen the rigor behind the claims (Supp Fig 1; Table S1). First, the synthesis argument obscures the relevant difference between sat mutagenesis & barcoded oPools. No one outside of a handful of academic labs and in industry would consider gene synthesis at scale like this. Second, the cost comparison presented here omits time and labor requirements of the method, which appear to be quite substantial. Specifically, the example gene PDR1 requires 43 sub-libraries to be constructed, each of which requires long-range PCR amplification of the original wild-type vector. These PCRs can be very tricky and often require multiple rounds of optimization of conditions. 43 of these reactions is quite a molecular biology load, and would scale unfavorably as you increase the length of YFG. Third, the claims behind the numbers are not well sourced and cannot be independently verified. In my opinion, the quality of the paper does not rest on an economic argument. In such cases, it's best to remove a cost comparison.

4. What is claimed in the paper (a method for size independent barcoded sat mut library generation) vs. what is demonstrated (barcoded libraries for two short tiles) is not reflected in the title, abstract, or introduction. Accurate description of what is demonstrated in the method, along with a discussion of strengths and limitations (n PCRs, n synthesized gene fragments of YFG, n parallel Gibson Assemblies, n parallel Golden Gate assemblies) in the discussion would improve the readability of the manuscript. Specifically, the authors did not demonstrate the method on multiple genes; did not create a barcoded full length saturation NNK library of their gene of interest; and did not perform a deep mutational scan on the full-length gene. While I don't think these experiments are necessary for publication of the method, they are necessary if the abstract and title remain unchanged.

Smaller comments

5. Figure 1C and 1D: it appears that the number of barcodes per variant is clipped at 10. This should either be mentioned explicitly, or better the raw values should be shown (this is better because the reader would get a better sense of the uniformity of the generated libraries).

6. Figure 2C: when evaluating the effect of number of barcodes on replicate correlation; was the number of short-read sequencing reads accounted for? Variants with higher number of barcodes will also tend to be more abundant in the library and their fitness will be better estimated simply due to increase short read coverage.

7. Figure 2 - what is the internal reproducibility of the assay (barcode vs. barcode or tile vs. tile)? And, how does this compare with the Figure 2 graph shown comparing barcode vs. tile?

8. The barcoding scheme uses position specific sequences along with degenerate sequences to tag each variant at a given position. Do the authors have any evidence that this scheme leads to higher fidelity variant-barcode associations compared with fully random barcodes?

Reviewer #2 (Olivier Tenaillon, signs review): In this manuscript by Jessica Jann entitled "Making deep mutational scanning accessible: a cost-efficient approach to construct barcoded libraries for genes of any length" propose a cost efficient method to do DMS in long genes with an coupling with Barcode.

The method is astute and straight forward and propose an intermediate coupling with barcode, which is truly relevant, as altenrative methods require long reads or emulsion PCR. Here the match between barcodes and sequence is extremely good.

I think the method will be useful for the community, including our lab.

I have just very minor comments:

twice it is mentioned:

-"multiple reactions and transformations were performed for each fragment" to reach the good numbers... It would be good to give a few examples of what multiple typically means... 3, 6, 20?

-The association from barcode to sequence is in theory imposed in the design. Here, a careful step of sequencing is done to validate it. Could it be possible to know the amount of differences between the initial coupling and the one observed. In other words, what would be the error rate if this "association sequencing" was not done and we jsut trusted the oPools sequences.

Overall it is a nice experimental protocol, that will be valuable for the community. Some, may be cheaper; alternative may exist but they do not include the coupling to barcodes, and end up having very large fraction of wild type sequences, which in the end make them much less cost competitive.

Olivier Tenaillon, I always sign my reviews.

Reviewer #3 (Guillaume Cambray, identifies himself): The manuscript "Making deep mutational scanning accessible: a cost-efficient approach to construct barcoded libraries for genes of any length" by Jann et al. describe an ingenious method to generate high-coverage variants libraries from relatively small pool of synthetic oligonucleotides. The authors present a proof-of-concept application, screening variant of yeast's PDR1 protein for increased antifungal resistance.

The method consists in using pools of degenerate oligonucleotides associating NNK random variations at each codon with partially random barcodes to generate trackable diversity through subsequent barcode-to-mutation mapping, and barcode frequency analysis upon functional screening. This enables substantial economy of scale in term of the number of oligonucleotides to synthetize. The authors further describe a two-step cloning strategy that enable targeting sequences that exceed the size of available oligonucleotide pools. Combined, these two aspects provide a convincing method to cost-efficiently generate large libraries of mutants. Although the authors focus on protein coding sequences, variations of the approach could be extensible to any kind of sequences of interest.

The method is conceptually sound and its efficiency demonstrated experimentally for two small and separate regions of PDR1. In my opinion, the manuscript nevertheless fails to demonstrate the applicability of the method to "genes of any length", as claimed in the title and repeatedly in the text. In effect, the manuscript only provides data for one sub-library covering 75 nucleotides of a 3,204-nucleotide gene (and in a more limited manner on another sub-library of 54 nucleotides).

While the core concept is appropriately demonstrated and it is implicitly stated that it can be easily scaled up, I do not think it does. I happen to be working with a similar strategy for constructing variants of a 6 kb genome—which yielded 45 sub-libraries—and I know for a fact that mixing these sub-libraries into a full library to perform pervasive screens leads to all sort of unforeseen issues. I therefore recommend that the manuscript demonstrate application to a sizable sequence length to substantiate one of the main claims of the paper. This might not need to be on the full extent of the pdr1 gene, but at least on a substantial length that exceed pure synthesis capacity (say 1kb).

I understand and respect that the authors might want to valorize their perhaps more extensive work with several publications, with the present manuscript being more method-oriented. I seem to understand that the authors chose to focus on a well-known region where mutants are already known and want to keep extensive descriptions of screening results on other regions for another functionally-oriented paper. There might be ways to present these data without focusing on (or even disclosing) their biological meaning, just showing that many sub-libraries can be constructed, pooled sufficiently uniformly to be assayed together and quality functional data be obtained over large sequences.

This being a method paper, I would also recommend to strengthen the precision of the materials and methods section.

I have listed below a number of additional comments that may help to improve the manuscript:

Figure 1:

Panel A: some aspect of the construction scheme could be depicted with more precision, e.g. including elements that are brought by oligonucleotides during PCR on the oligos. Some elements are also at odd with the description on the metod section (see details below)

Panel B: Reading the main text, it was particularly unclear to me where the data shown here came from. I seem to understand from the previous section that these are derived by cumulating sequencing sequencing data obtained separately on independent small-scale transformations. This should be stated clearly in the text. Points should be added to the graphs in addition to lines, since it is difficult to tell where the straight lines break.

The denomination ""% barcode diversity" was somehow difficult to grasp. I seems to understand that this quantity represents the percentage of sequence associated with at least 4 and at least 9 informative barcodes? Maybe something like #barcode>4 and #barcode>9 could be clearer (this also apply to panel CD).

l102: "Means and confidence intervals obtained for 100 random draws of transformation reactions": I am not sure I understand what this means exactly.

Panel CD: the thremomter should show discretized colors and the color scheme should make it easier to tell exactly how many barcodes are identified in each position. Data on the rightmost panel would be better shown as a barplot.

Figure 2:

Panel B: I understand that a heatmap conveys more information per se, but comparing data from direct and barcode-based mutant identification would be better conveyed with a scatter plot of the one vs the other. Points could be additionally colored by the number of barcodes from which the values are estimated for the barcode-base identification. The heatmap for direct sequencing is not necessary and could either be removed or kept is space allows.

More generally, an explanation of the few observed discrepancies between the two approaches could be discussed.

The NNK strategy can yield one stop codon at each position. The heatmap from direct sequencing show occasional positive selection for these, while that from direct identification show mildly negative values. I would expect highly negative selection for these mutations and think that the observed data deserve some explanation: is it expected that truncated protein remain functional? Is it due to leakage from the endogenous gene that is normally dominated by the plasmid-borne variants? Is it indicative of a lack of measurement precision?

Panel C: l204-205: "Means and confidence intervals were calculated from 100 random subsamplings per variant." I do not see error bars here. Also, it is unclear how these data are generated: are variants with more or less than N barcodes left out from the correlation? If so, the correlation obtained by using all usable data should be communicated.

l148: "The performance of a DMS plasmid library depends on three key parameters": I do agree with all these parameters but would argue that a fourth parameter quantifying representation bias between variants is also key. Such bias can come from synthesis, construction, subsequent growth, etc… Significant distribution biases hinder cost-effectivess as much sequencing throughput is wasted on the most abundant to get data for the least abundant variants. A metric should be chosen to quantify this.

l218-219: "estimates from the barcodes or from the F13 coding region were strongly correlated (r = 0.95, 10 barcodes)": it would be better to highlight the correlation when all available barcodes are used.

l342-343: "Since oPools are synthesized in the 5' to 3' direction, this initial step minimizes the incorporation of truncated oligos (linked to oPool incomplete synthesis) into the final product": this is factually wrong, the phosphoramidite chemistry used for all chemical oligonucleotide synthesis actually proceeds in the 3' to 5' direction. This first PCR and associated purification seems rather useless.

l348-350: "This approach improved the overall quality of the oPools, ensuring sufficient insert quantity and diversity for subsequent steps of DMS library construction.": I don't think this claim is substantiated by data, please edit to something along the line of "This approach is intended to improve..."

l357-359: "The resulting PCR product was incubated for 1h30 at 37 °C with 10 U of DpnI enzyme to remove parental plasmid DNA and then, purified using magnetic beads.": In this description the linear PCR product is directly used for Gibson assembly. This is not what is suggested in Figure 1A, where it appear as a circular intermediate obtained from transformation (which would have been better to avoid PCR-related mutation in the backbone, including part of YFV). Likewise for the second cloning step, the missing fragment would have better been cloned along with flanking BsaI and sequence verified before used in golden-gate. I suggest the author add such recommendations to the method section as potential improvements.

l363-365: "Following transformations, 5 mL of 2YT medium was added directly onto each agar plate": this is too imprecise for a methodological paper. What transformation procedure is used (chemical, electroporation)? As I understand 2YT was added to the cell after the transformations. Were cell grown after that before plating. At what point are transformant counted?

l380-381: "To achieve this, multiple parallel Golden Gate reactions and transformations were performed for each fragment." This is surprising to me, a transformation efficiency of 10^5 is fairly easy to attain, especially with commercial competent cells...

l 389-… Plasmid library quality control by sequencing: Description of the the sequencing libraries preparation comes after description of sequencing read processing. It would be clearer to have a dedicated section for the various flavors of sequencing library preparation.

More broadly, amplicon sequencing is notoriously biased, with random molecules sometime getting way more amplified than other, which is a big problem for downstream quantitative analyses. A now common approach to mitigate this is to incorporate Unique Molecular Identifier during PCR. The authors have not done this here, but could discuss this as a weakness and suggested it as an improvement.

l446-....: "Variant library construction": most of this description is not matched by any main text. A few main text sentences describing that PDR1 is essential, has been placed under a dox-repressible promoter to enable turning its expression down and conditional complementation by a plasmid library would be welcome.

l535-536: "an initial PCR was carried out to amplify only the PDR1 from the plasmid": I understand that this is necessary for the direct sequencing of F13 variants, and it makes sense to use the same material for barcode library prep in the framework of comparing data derived from barcode sequencing and direct variant region sequencing. However, higher quality data on barcodes could be derived by directly targeting PBS_i5 and PBS_i7 priming site...

l563-...: "Selection coefficient and characterization of the variants". The authors have already established a pipeline to identify barcodes. As the very same pipeline can presumably be used to count barcodes, what exactly is the added value of using gyoza?

l577: "normalization by the number of mitotic generations": this needs to be explained.

l578: "subtraction of the median log2 fold-change of silent variant": likewise, this require additional explanation. What is the fold-change distribution of silent variant, where is the actual WT in this distribution? How does it compare with the distribution of variants?

l581-583: "Variants were assigned confidence scores based on their initial abundance at TP0 across replicates. A minimum threshold of five reads per mutation was used to define high-confidence variants": Does this mean that all replicates must have at least 5 reads? As far as I can tell, there has been no mention of experimental replicates until this point: what are they?

As much as the number of initial reads is important, this seems a little light to qualify as a confidence score... I guess the spread of values around barcode replicates and synonymous amino-acids could be used for that purpose, but neither is mentioned, despite means being considered (l584).

Finally, a potential weakness of the strategy could be discussed. In the construction scheme, each codon position is associated with a defined barcode core of 24 nts, while 6 barcode positions are degenerate. One can't really rule out a potential functional effect of the core barcode, which would then systematically impact all variants at that position. Are there ways to diagnose or mitigate such effects?

Reviewer #4: Systematic mutational scanning is a powerful technique that enables researchers to assess the functional impact of all possible point mutations in a gene within a single experiment, providing a comprehensive view of the relationship between protein sequence and function. However, generating libraries for these experiments is often prohibitively expensive and labor-intensive, with costs increasing proportionally to gene size.

The manuscript by Jann et al. introduces a cost-effective and accessible method construction of a barcoded library that addresses these challenges. Overall, the manuscript is clearly written and well-presented, and the authors' elegant approach to library design will be valuable for future DMS studies. I have just a few comments and questions that arose while reading the manuscript. Overall, I recommend this manuscript for publication in PLOS Biology.

1. It may be helpful to include a table summarizing coverage statistics for each of the two library fragments constructed in this study. This could include the number of transformants for the two cloning steps, sequencing reads, informative barcodes, mutation coverage, barcode diversity, and other relevant metrics.

2. While the oPool-based method is clearly more affordable than purchasing a commercial library, it appears potentially labor-intensive—particularly when synthesizing a large gene such as PDR1 in 43 separate fragments. Could the authors provide an estimate of how many days it would take to clone the entire PDR1 library using this strategy?

3. In Supplementary Figure 1A, the cost of cloning a commercially synthesized library is estimated at approximately $60 per amino acid position. I was curious how this number was calculated? Based on a rough back of the envelope estimate—assuming ~100,000 transformants per Gibson assembly reaction and thus approximately 10 reactions to cover a 300-aa barcoded library—the total cost in my mind would only be a few hundred dollars in total. Could the authors clarify the basis of the $60/aa cost?

---

## [Decision Letter · Decision Letter 2]

5 Jan 2026

Dear Christian,

Thank you for your continued patience while we considered your revised manuscript "Making deep mutational scanning accessible: a cost-efficient approach to construct barcoded libraries for genes of any length" for publication as a Methods and Resources Article at PLOS Biology. Please accept my sincere apologies for the delays that you have experienced during this round of the peer review process. This revised version of your manuscript has been evaluated by the PLOS Biology editors, the Academic Editor and two of the original reviewers. In addition, I have provided some specific remaining comments from the Academic Editor below the reviewer reports (labelled 'Comments from the Academic Editor').

Based on the reviews, I am pleased to say that we are likely to accept this manuscript for publication, provided you satisfactorily address the remaining points raised by Reviewer's #1 and #3. After discussions with the academic editor, we will not make the request to include additional validation for the screening of pooled sub-libraries essential for the revision, but we encourage you to include this data if you have it to hand.

In addition, please make sure to address the following data and other policy-related requests that I have provided below (A-G):

(A) We routinely suggest changes to titles to ensure maximum accessibility for a broad, non-specialist readership. In this case, we would suggest the following edit to the title. Please ensure you change both the manuscript file and the online submission system, as they need to match for final acceptance:

“A cost-effective and scalable barcoded library construction method for deep mutational scanning studies”

(B) We note that the Abstract is currently 71 words and we would be grateful if it could be expanded at this stage to around 150 words to fully contextualize and summarize the methodological developments offered by the study.

(C) You may be aware of the PLOS Data Policy, which requires that all data be made available without restriction: http://journals.plos.org/plosbiology/s/data-availability. For more information, please also see this editorial: http://dx.doi.org/10.1371/journal.pbio.1001797

-Supplementary files (e.g., excel). Please ensure that all data files are uploaded as 'Supporting Information' and are invariably referred to (in the manuscript, figure legends, and the Description field when uploading your files) using the following format verbatim: S1 Data, S2 Data, etc. Multiple panels of a single or even several figures can be included as multiple sheets in one excel file that is saved using exactly the following convention: S1_Data.xlsx (using an underscore).

-Deposition in a publicly available repository. Please also provide the accession code or a reviewer link so that we may view your data before publication.

Figure 1B-D, 2B-C, S1, S2, S3, S4, S5B, S6, S9

(D) I appreciate that the data requested above may be contained within your Github deposition (https://github.com/Landrylab/Jann_et_al_2025). If so, I would be grateful if you could clearly label where the underlying data can be found in the deposition folders.

(E) Please note that we cannot accept sole deposition of code in GitHub, as this could be changed after publication. However, you can archive this version of your publicly available GitHub code to Zenodo. Once you do this, it will generate a DOI number, which you will need to provide in the Data Accessibility Statement (you are welcome to also provide the GitHub access information). See the process for doing this here: https://docs.github.com/en/repositories/archiving-a-github-repository/referencing-and-citing-content

(F) Please also ensure that each of the relevant figure legends in your manuscript include information on *WHERE THE UNDERLYING DATA CAN BE FOUND*, and ensure your supplemental data file/s has a legend.

(G) Please ensure that your Data Statement in the submission system accurately describes where your data can be found and is in final format, as it will be published as written there.

(H) Please note that per journal policy, the species studied should be clearly stated in the abstract of your manuscript.

We expect to receive your revised manuscript within three weeks.

*Published Peer Review History*

*Press*

Best regards,

Richard

Richard Hodge, PhD

rhodge@plos.org

Reviewer remarks:

Reviewer #1: The authors have addressed most of my comments; they have added new experimental data scanning the entire gene. One very small note - in Figure 2B the PDR1 name is clipped by a white opaque box in the updated figure; this can be handled in the proof.

Reviewer #3 (Guillaume Cambray, identifies himself): This revised version of the manuscript "Making deep mutational scanning accessible: a cost-efficient approach to construct barcoded libraries for genes of any length" by Jessica Jann et al. largely addresses most of my previous questions and concerns, as well as those raised by the other reviewers.

A major comment—also raised by another reviewer—concerned the manuscript's emphasis on the gene-scale applicability of the approach, while only two short regions were presented. The authors have now provided library construction data for all 43 regions of the gene (Sup. Figs. 7 and 8), which addresses this comment only in part.

The innovative aspect of this work lies in coupling NNK mutagenesis of a relatively short region (i.e., one that can be sequenced using widely available and cost-efficient short-read sequencing such as PE150) with a partially randomized barcode. As the authors point out, different barcodes attached to the same mutation can be viewed as internal replicates that enhance measurement accuracy and statistical robustness. Nevertheless, direct sequencing of the mutated region remains the gold standard, as used by the authors in Figure 2BC and Sup. Figs. 6 and 9.

A major advantage of the barcode strategy is that multiple sub-libraries can be pooled and assayed simultaneously rather than separately, effectively performing one large experiment instead of 43 independent ones. Beyond the obvious cost efficiency, this pooling also improves data consistency and reduces the need for normalization. In my view, a complete validation of the proposed approach should demonstrate that a pool of sub-libraries can indeed be successfully screened. Without pooling, barcoding brings little added value: a single cloning of an NNK-mutagenized oPool, followed by sequencing of the mutated region—as performed here and in other studies (Tile-Seq)—would be sufficient.

In their response, the authors suggest that because they have successfully pooled sub-libraries previously, there is no need to demonstrate the effectiveness of pooling here. By that logic, however, there would have been no need to perform a screening experiment on a single barcoded sub-library, since barcoding itself is also a previously established strategy.

I emphasize this point because, in a very similar experimental context, pooling has unexpectedly caused issues in our own hands—even though we had previously pooled complex libraries successfully. This might be specific to our system, but it would be unfortunate if this manuscript did not address pooling experimentally to fully substantiate the method. I am not asking for a full gene-scale pool, but rather a small number of pooled regions (preferably contiguous, to highlight continuity in measurements across regions). This seems feasible considering all sub-libraries have already been constructed. Again, the goal is not an extended functional analysis—which could be published separately—but a demonstration that substantiates the method's core claim.

Aside from this major point, I have a few remaining comments and suggestions, detailed below:

* Lines 113-114:

"Means and confidence intervals obtained for 100 draws of randomly picked transformants."

Despite the revision, this statement remains unclear. To my understanding, a transformant refers to a single clone. The description in the Methods section is clearer. I suggest rephrasing along the lines of: "100 ordered arrangements of independent transformation experiments with 5,000 and 7,500 clones, respectively, were randomly drawn to construct this graph."

* Figure 1B:

Unless I am missing something, I do not understand what the statistics would indicate when the number of transformants per base pair is 0. I believe there may be an error in the x-axis labeling.

* Lines 198-200:

"~350 transformants per base pair are needed to optimize sufficient mutation coverage, barcode diversity, and informative barcodes (Figure 1B)."

This seems to apply only to F13. In contrast, F43 suggests that ~150 transformants would suffice. This sizable difference deserves discussion. What accounts for the discrepancy? Were similar analyses performed for other regions? If so, do additional patterns emerge? I am also uncertain how many transformants per bp were finally used for the functional screening of Figure 2.

* Figures 1C-D:

I had not realized previously (though it now seems obvious) that the number of barcodes was clipped at 10. As Reviewer 1 suggested, the non-clipped versions are now shown in Sup. Figs. 1 and 2. In my opinion, these non-clipped figures should appear in the main text, as the clipped versions obscure important information. The observation that only a few mutations have fewer than 5 barcodes can instead be described in the text, with reference to the clipped data as supplementary information (or very low barcode counts could be color-coded differently in the main figure).

The new Sup. Figs. 1 and 2 make it clear that systematic biases exist in barcode coverage. Much of this can be attributed to the number of synonymous codons generated by NNK mutagenesis. For example, Leucine, Arginine, and Serine have more barcodes because they are encoded by three possible codons. Valine, Alanine, Glycine, and Proline each have two codons, yet Proline and Alanine appear underrepresented in both F13 and F43. Since Sup. Figs. 7 and 8 are also clipped at 10, it is difficult to assess whether this pattern generalizes more broadly. I have no personal experience with NNK mutagenesis—perhaps these biases are known—but they merit some discussion, as they represent a minor limitation compared with more expensive strategies in which amino acid substitutions are preprogrammed in the oligo sequences.

COMMENTS FROM THE ACADEMIC EDITOR

- The new Figure 2C is almost unreadable; at least the text size has to be increased.

- All code should be archived (e.g., on Zenodo) - the github links should be replaced with dois to the archived versions.

- In Figure 1 C and D it should say ">=10" in the heatmap legend.

- In the Supplement, I would have appreciated an actual distribution plot of the barcode numbers in addition to the heatmaps and the uniformity coefficients.

---

## [Editor Report · Decision Letter 3]

26 Jan 2026

Dear Christian,

On behalf of my colleagues and the Academic Editor, Claudia Bank, I am pleased to say that we can accept your manuscript for publication, provided you address any remaining formatting and reporting issues. These will be detailed in an email you should receive within 2-3 business days from our colleagues in the journal operations team; no action is required from you until then. Please note that we will not be able to formally accept your manuscript and schedule it for publication until you have completed any requested changes.

PRESS

Best wishes,

Richard

Richard Hodge, PhD

rhodge@plos.org

PLOS
